# A novel framework to investigate wind-driven snow redistribution over an Alpine Glacier: Combination of high-resolution Terrestrial Laser Scans and Large-eddy simulations

Annelies Voordendag[1,2,*], Brigitta Goger[1,3,*], Rainer Prinz[1], Tobias Sauter[4], Thomas Mölg[5], Manuel Saigger[5], and Georg Kaser[1]

[*]These authors contributed equally to this work.
[1]Department of Atmospheric and Cryospheric Sciences, Universität Innsbruck, Innsbruck, Austria
[2]Institute of Geodesy and Photogrammetry, ETH Zurich, Zurich, Switzerland
[3]Center for Climate Systems Modeling, ETH Zurich, Zurich, Switzerland
[4]Geographisches Institut, Humboldt-Universität zu Berlin, Berlin, Germany
[5]Climate System Research Group, Institute of Geography, Friedrich-Alexander-Universität (FAU) Erlangen-Nürnberg, Erlangen, Germany

**Correspondence:** Annelies Voordendag (annelies.voordendag@geod.baug.ethz.ch), Brigitta Goger (brigitta.goger@c2sm.ethz.ch)

**Abstract.** Wind-driven snow redistribution affects the glacier mass balance by eroding or depositing mass from or to different parts of the glacier's surface. High-resolution observations are used to test the ability of large eddy simulations as a tool for distributed mass balance modeling. We present a case study of observed and simulated snow redistribution over Hintereisferner glacier (Ötztal Alps, Austria) between 6 and 9 February 2021. Observations consist of three high-resolution Digital Elevation Models ($\Delta x$=1 m) derived from terrestrial laser scans taken shortly before, directly after, and 15 hours after snowfall. The scans are complemented by data sets from three onsite weather stations. After the snow fall event, we observed a snowpack decrease of 0.08 m on average over the glacier. The decrease of the snow depth can be attributed to post-snowfall compaction and wind-driven redistribution of snow. Simulations were performed with the WRF model at $\Delta x$=48 m with a newly implemented snow drift module. The spatial patterns of the simulated snow redistribution agree well with the observed generalized patterns. Snow redistribution contributed -0.026 m to the surface elevation decrease over the glacier surface on 8 Feb, resulting in a mass loss of -3.9 kg m$^{-2}$, which is in the same order of magnitude as the observations. With the single case study we cannot yet extrapolate to the impact of post-snowfall events on the seasonal glacier mass balance, but the study shows that the snow drift module in WRF is a powerful tool to improve knowledge on wind-driven snow redistribution patterns over glaciers.

## 1 Introduction

The European mountain cryosphere is an important contributor to Alpine water availability and experiences, as the worldwide cryosphere, the effects of global climate warming (e.g. Fox-Kemper et al., 2021; Hock et al., 2022). The annual mass balances of the Alps' glaciers are increasingly more negative since the 1980's (Marzeion et al., 2012; Huss and Hock, 2018; Hugonnet et al., 2021), and extreme glacier mass losses are observed in more recent years (Copernicus Climate Change Service (C3S),

2023; Voordendag et al., 2023b; Cremona et al., 2023). However, a knowledge gap still exists on the impact of small-scale pro-
cesses such as cryosphere-atmosphere exchange or wind-driven snow transport on snow accumulation over mountain glaciers
(e.g. Mott et al., 2018; Beniston et al., 2018). Spatial observations of snow cover changes on mountain glaciers are sparse and
often only available on the point scale, and numerical weather prediction models on the kilometric range are not able to resolve
the relevant small-scale boundary layer processes and surface fluxes over highly mountainous terrain (Vionnet et al., 2016;
Goger et al., 2018, 2019; Gouttevin et al., 2023). On the other hand, distributed mass balance models (Machguth et al., 2006),
e.g. COSIPY (Sauter et al., 2020), require high-resolution input fields to deliver respective information about a glacier's surface
mass balance. Among the usual meteorological variables (e.g., temperature, wind speed, relative humidity, total precipitation,
etc), snow depth can also be used as an initial condition, improving the accuracy of distributed surface mass balance models.

In general, snow depth distribution over complex terrain cannot be assumed to be homogeneous. The spatial precipitation
pattern over mountains is heterogeneous due to multi-scale interactions of the atmospheric flow with topography (Frei and
Schär, 1998; Isotta et al., 2013; Colle et al., 2013). Furthermore, during or after snowfall, the depth of the snowpack is affected
by four processes: melt, compaction, sublimation, and wind-driven snow redistribution. Compaction of the snowpack can be
driven by the overburden of its own weight, the pressure exerted by the wind and/or snow metamorphism processes. Snow
redistribution is the relocation of wind-borne snow, or also called snow drift, from one part of the snow-covered area to another
(Cogley et al., 2011; Mott et al., 2018). Redistributed snow leads to snow depth decrease in areas where snow is eroded, and
snow cover increases, where snow particles are deposited. The resulting snow patterns strongly depend on the local topography,
and the wind speed and direction (Gerber et al., 2017; Vionnet et al., 2013, 2021; Sauter et al., 2013). The complex terrain
makes mountain glaciers subject to heterogeneous snow cover distribution caused by both complex precipitation patterns and
wind driven redistribution during and after snow fall (Dadic et al., 2010).

It is still a challenge to measure the spatial (re)distribution of the snow cover continuously in a complex alpine environment.
One possible method to record glacier-wide snow distribution of precipitation and the post-snowfall surface elevation changes
over a glacier is with repeated Digital Elevation Models (DEM) derived from terrestrial or airborne laser scanning (TLS/ALS).
In recent times, surface elevation changes at mountain glaciers were measured with both TLS (Fischer et al., 2016; Prantl et al.,
2017; Xu et al., 2019; Mendoza et al., 2020) and ALS (Grünewald et al., 2014, Table 2). However, these DEMs are acquired
irregularly and at low temporal resolution. Long-term, continuous data series that capture snow fall and snow redistribution
over a certain area are not available so far. This gap was addressed with the installation of a permanent TLS station nearby
Hintereisferner (HEF) glacier, located in the Ötztal Alps, Austria (Voordendag et al., 2021b). This TLS station acquires a daily
DEM automatically, but even hourly acquisitions are possible if manually initiated. A comprehensive uncertainty assessment
shows that this TLS station is able to capture small glacier surface changes, such as snow (re)distribution (Voordendag et al.,
2023a). The high temporal and spatial data resolution contribute to improving the process-understanding at HEF and can be
used to evaluate. surface elevation changes in atmospheric model simulations. Modelling snow processes is usually achieved
by a large variety of standalone snow models, which receive input data from atmospheric models or observations (Krinner
et al., 2018; Menard et al., 2021). Recent studies also coupled full (previously) stand-alone snowpack models with atmospheric
models. For example, Vionnet et al. (2014) coupled the Crocus snow model with the Méso-NH LES model to explore snow ac-

cumulation patterns. They found that the wind-induced snow redistribution is responsible for an increase in spatial variability in snow depth. The most recent development in this direction is CRYOWRF (Sharma et al., 2023), where the SNOWPACK model (Lehning et al., 1999), including a snow drift module, was coupled to the Weather Research and Forecasting (WRF) model (Skamarock et al., 2019). First results suggest that CRYOWRF is capable of simulating snow accumulation and redistribution over the Swiss Alps and Antarctica (Sharma et al., 2023; Gerber et al., 2023).

While fully coupled snow-atmosphere model chains likely resolve coupled processes and atmosphere-cryosphere interactions well for case studies (at a high numerical cost), common numerical weather prediction (NWP) models include multi-layer snow schemes within their land-surface models, e.g. the NOAH-MP scheme in the WRF model (Niu et al., 2011) or the snow model in the Integrated Forecasting System (IFS, Arduini et al., 2019). Usually, these land surface models are less complex than full snow models and do not include a package for wind-driven snow redistribution, although the horizontal resolution of NWP models keeps decreasing and process studies at large-eddy simulation (LES) resolution ($\Delta x \approx \mathcal{O}(10m)$) over mountainous terrain became more and more relevant for process understanding in the recent years (e.g., Gerber et al., 2018; Umek et al., 2021; Goger et al., 2022). At this resolution, both topography and glacier ice surfaces in the Alps can be expected to be well-resolved, given that at least 10 grid points across a valley are necessary to resolve the relevant boundary-layer processes (Wagner et al., 2014). This criterion is clearly met over HEF in the summer glacier boundary layer simulations at $\Delta x =$48 m by Goger et al. (2022).

Recently, the `snow2blow` snow scheme by Sauter et al. (2013) was implemented in the WRF model by Saigger et al. (2023). To our current knowledge, this is the first time where an openly available, easy-to-use (i.e., no changes in compilation procedure etc.) formulation for wind-driven snow redistribution is implemented in the WRF model code. In this study, we combine the high-resolution LES setup by Goger et al. (2022) with the TLS scans from Voordendag et al. (2021b) to study the impact of wind driven snow redistribution on a large Alpine glacier for a case study. We present a first evaluation of the newly implemented snow drift scheme with high-resolution TLS observations and examine whether the model delivers realistic results in snow depth change and spatial patterns in this highly complex environment. Furthermore, with the aid of the model, we can also try to disentangle the physical processes affecting the snowpack on the glacier, which cannot be determined via the surface elevation changes measured by the TLS. Finally, we can give a cautious estimate on the impact of wind-driven snow redistribution on glacier mass balance.

This paper is organised as follows: First, we describe our study area and the selection of the case study between 6 and 9 February 2021. In Section 2 we give an overview of the TLS station, meteorological observations, and the model setup with the implemented snow drift module. The first part of the results (Sect. 3) includes the observed snow depth changes with the TLS, an overview of the meteorological situation at the glacier as seen by point observations, and evaluation of the model performance in terms of precipitation, wind patterns, and snow water equivalent changes. In the second part of the results, the TLS data is compared with the model output on wind-driven snow redistribution. We estimate the snow compaction from the observational data and give a final assessment on the reliability of the model data. We deliver a detailed discussion on the advantages and shortcomings of our setup (Sect. 4). Finally, we conclude and discuss the future implications of this work.

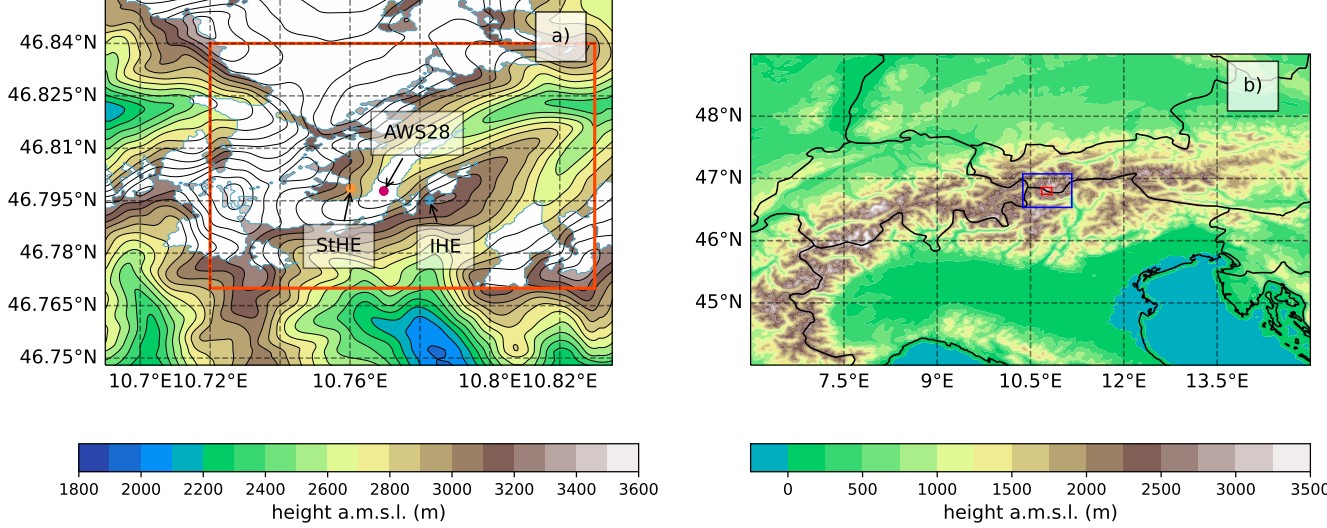

**Figure 1.** a) Overview of the innermost model domain ($\Delta x = 48\,\mathrm{m}$, red rectangle) with the model topography (contour lines) and the glacier areas as represented in the model (white area with light blue outlines). The locations of the three stations StHE, AWS28, and IHE are highlighted in colors. b) The mesoscale domain ($\Delta x = 1\,\mathrm{km}$) spanning the Alps with the two LES domains highlighted in blue ($\Delta x = 240\,\mathrm{m}$) and red ($\Delta x = 48\,\mathrm{m}$).

## 2  Methods

### 2.1  Study area and available observations

The Hintereisferner (HEF) is a large valley glacier located in the Ötztal Alps, Austria (Figs. 1a, 2). HEF is a principal research site to study glaciological processes since the early days of glacier research (Blümcke and Hess, 1899). Annual and seasonal glaciological mass balance measurements are acquired since 1952/53, and HEF is classified as one of the 'reference glaciers' by the World Glacier Monitoring Service (WGMS, Zemp et al., 2009). HEF has a length of approximately 6.3 km and stretches from its highest point, the Weißkugel (3738 m a.s.l.), down to a terminus altitude of 2460 m a.s.l. (data from 2018).

The glacier and its surroundings are well-equipped with several automated weather stations as part of the Rofental catchment observational network (Strasser et al., 2018). The major station for this study is *Im Hinteren Eis* (IHE), exists since 2016 and is located on the orographic right side of the glacier on the ridge at the Austrian-Italian border (Fig. 1). IHE is equipped with a permanently installed TLS device, which is extensively described in Voordendag et al. (2021b, 2023a) and Sect. 2.3 of this study. Additionally, two webcams, which deliver images every thirty minutes[1], are installed at the position of the TLS and

overlook the glacier surface. About 50 m from the container with the TLS, an eddy-covariance flux tower is installed at the mountain ridge (Table 1) providing turbulence observations which have been used for fundamental evaluation of boundary-layer theory (Stiperski et al., 2021; Stiperski and Calaf, 2023) and model evaluation (Goger et al., 2022). After post-processing

---

[1]www.foto-webcam.eu/webcam/hintereisferner1/

|  | Im Hinteren Eis | Station Hintereis | AWS28 |
|---|---|---|---|
| Latitude | 46.795761 | 46.798896 | 46.79779 |
| Longitude | 10.783409 | 10.760373 | 10.76967 |
| Altitude (m a.s.l.) | 3264 | 3031 | 2782 |
| Air pressure | Setra278 CS100 | Setra278 CS100 | Vaisala PTB 110 |
| Air temperature | Rotronic HC2-S3 Ventilated; at 1.50 m and 5.50 m | Campbell Scientific EE181 Naturally ventilated; at 2.10 m | Campbell Scientific CS215 |
| Precipitation | Geonor T200B | Ott Pluvio2L | |
| Radiation | Kipp&Zonen CNR4 | Kipp&Zonen CNR4 | Kipp&Zonen CNR4 |
| Relative humidity | Rotronic HC2-S3 Ventilated; at 1.50 m and 5.50 m | Campbell Scientific EE181 Naturally ventilated; at 2.10 m | Campbell Scientific CS215 |
| Snow depth | Campbell Scientific SR50ATH-L | Campbell Scientific SR50A | Campbell Scientific SR50A |
| Wind speed and direction | Lufft Ventus-UMB At 1.50 m, 3.00 m and 6.00 m | Young 05103-45 Alpine At 3.30 m | Young 05103 |
| 3D sonic anemometer | Metek uSonic-3: At 3.18 m | | |
| TLS | RIEGL VZ-6000 | | |
| Webcams | Canon EOS1200D (2x) | | |

**Table 1.** Location, altitude and available observations at Im Hinteren Eis, Station Hintereis and AWS28 during the period of the case study.

(Stiperski and Rotach, 2016; Rotach et al., 2017), the averaged variables (e.g., wind speeds, air temperature, surface fluxes, and turbulence kinetic energy) are available at a 15-minute interval.

The second station is *Station Hintereis* (StHE, Fig. 1a), located on the orographic left side of the glacier equipped with an automatic weather station (Table 1) and a mountain hut used for logistical support. At this location, meteorological measurements were conducted for more than 50 years (Obleitner, 1994), mostly during the summer season. Continuous, all-season observations of common meteorological variables are available since 2010.

Last, a temporary automatic weather station was installed at the glacier in the line of sight between IHE and StHE from 7 Dec, 110 2020 to 22 Feb, 2021, and will be called *AWS28* hereafter, as it was installed at an altitude of approx. 2800 m a.s.l.. It provides common meteorological measurements (Table 1). The meteorological observations from the three stations are used to explain the meteorological situation and to validate the simulations (see Sect. 2.4) on a point scale.

## 2.2 Case Study Selection

The case study has been selected based on the meteorological observations and the images recorded by the webcams by 115 applying the following criteria:

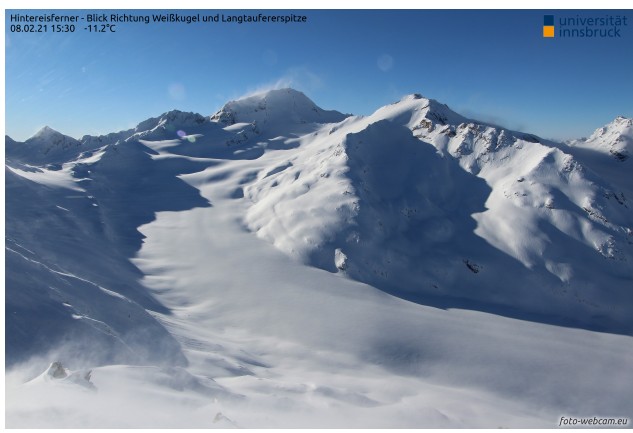

**Figure 2.** Webcam image of 8 Feb, 2021, 15:30 (UTC+1) showing signs of snow drift at the mountain ridges. The image is retrieved from https://www.foto-webcam.eu/webcam/hintereisferner1/2021/02/08/1530.

- the period should be in winter and show a pronounced change of the synoptic weather situation with a subsequent accumulation of fresh snow;

- wind speeds above $5\,\mathrm{m\,s^{-1}}$ should be observed during or directly after the snowfall event to ensure wind-driven snow redistribution;

- no surface elevation change due to melt should be occurring;

- frequent TLS scans must be available (Sect. 2.3).

The time window of 6 Feb-9 Feb, 2021 met these criteria.

The large-scale synoptic situation from ERA5 reanalysis data revealed that the Alps were under the influence of large-scale Southerly flow and moisture transport from the Mediterranean Sea. The southerly flow was mostly associated with a trough over France moving eastward towards the Alps, while the trough axis passed our location of interest on 7 Feb 2021 after 18:00 UTC. The associated surface frontal system brought pre-frontal snow fall, which ceased in the early morning hours of 8 Feb, while the actual cold front passed the glacier likely around 09:00 UTC, together with a rise in air temperatures and a decrease in cloud cover. After the trough passage, winds at upper levels shift towards Westerlies, while at crest levels, winds shift to Westerlies to South-westerlies. At around 12:00 UTC, the wind speeds increased to over $5\,\mathrm{m\,s^{-1}}$, providing excellent snow drift conditions. Webcam imagery (Fig. 2) of 8 Feb, 15:30 local time (UTC+1) shows blowing snow at the mountain ridges surrounding the glacier, indicating high wind speeds and snow drift.

## 2.3 Terrestrial laser scanning acquisitions

The location IHE is equipped with a permanently installed and automated TLS station (Voordendag et al., 2021b). The TLS station is in operational daily use since 2020, and thus delivers a daily point cloud of HEF under clear weather conditions

(e.g. no clouds between TLS and target surface). The TLS station is normally set to a daily acquisition at 01:42 UTC, but as the end of a snowfall period was observed on the webcam images on 8 Feb, 10:22 UTC, an additional scan acquisition was initialized. This led to three usable scans for the case study: shortly before the snowfall event on 6 Feb (01:42 UTC), directly after the snowfall 8 Feb (10:22 UTC) and approximately 15 hours after snowfall ended on 9 Feb (01:42 UTC). In the following text, we refer to these scans as scan 1, 2, and 3, respectively. The acquired point clouds are registered to each

other with the RiSCAN PRO software (RIEGL, 2019) and gridded to Digital Elevation Models (DEMs) with an one-meter horizontal resolution. Voordendag et al. (2023a) investigated surface change processes that can be captured by the TLS station. They found that the scans have an uncertainty of $\pm0.10$ m in vertical direction after the registration in RiSCAN PRO. In this study, the scans were registered with manually selected tie objects, such as snow-free rocks and the walls of StHE, which led to a better registration than the calculated $\pm0.10$ m in vertical direction with automatically selected tie planes in Voordendag

et al. (2023a). Additionally to the one-meter grid size DEMs, the high-resolution point clouds are gridded to DEMs with a $\Delta x = 48$ m allowing a direct comparison to the numerical simulations.

## 2.4   Numerical model

We employ the Weather Research and Forecasting (WRF) model version 4.1 (Skamarock et al., 2019) in a nested setup for the numerical simulations of the case study period. The numerical setup is the same as described by Goger et al. (2022), therefore

we only mention the most relevant aspects. As model boundary conditions, we use ERA5 reanalysis data, feeding the outermost WRF domain ($\Delta x = 6$ km) spanning Europe, and subsequently nesting down across $\Delta x = 1$ km (mesoscale domain) to the two LES domains at $\Delta x = 240$ m and $\Delta x = 48$ m, respectively (Fig. 1). We choose the Shuttle Radar Topography Mission (SRTM) 1 Arc-Second Global topography dataset (USGS, 2000) as raw data for the model topography. Due to numerical constraints, the topography data has to be terrain smoothed with a 1–2–1 smoothing filter (Guo and Chen, 1994). For the two domains

at the hectometric range, the coarser domain's topography was interpolated to the grid of the higher-resolution domains, and slopes steeper than 30° were replaced with slopes from the respective coarser-grid topography to avoid numerical instabilities. Henceforth, instead of applying more smoothing cycles, we can keep a part of terrain heterogeneity with this method. We utilize ESA-CCI land cover (ESA, 2017) for the two outer domains, while we put a special focus on the correct representation of land-use and glacier outlines in the LES domains. We use the CORINE land-use data set (European Environmental Agency,

2017) with an additional correction of the ice surfaces of the glaciers as described in Goger et al. (2022). NOAH-MP (Niu et al., 2011) is used as a land-surface scheme, which includes a three-layer snow model. Snow compaction by the snowpack's own weight is calculated following the empirical relations by Anderson (1976) and Sun et al. (1999). Furthermore, we implemented a novel snow drift module as described in Sect. 2.4.1. We use the Thompson microphysics (Thompson et al., 2008), where snow assumes a nonspherical shape with a bulk density varying with diameter. The MM5 revised surface layer scheme (Jiménez et al.,

2012), and the RRTMG two-stream radiation scheme (Iacono et al., 2008) with topographic shading for all domains are utilized. For the boundary layer turbulence we employ the MYNN parameterization (Nakanishi and Niino, 2009) for the two outermost domains, while we switch it off in the LES domains and employ the turbulence closure after Deardorff (1980). Furthermore, we also use the online averaging module by Umek et al. (2021), therefore, all model output shown in the following are 15-minute

averages. The model is initialized on 8 Feb, 00:00 UTC, and runs for 24 hours. We would like to stress that the snowpack in the model is initialized at the same time. This might introduce a slight bias in snowpack density, since the model initializes the snowpack as "fresh snow", while in reality, an older snowpack is already present at the glacier and its surroundings. However, due to the expensiveness of the LES, a long spin-up period of e.g., weeks, is not feasible with our current setup. We consider this possible shortcoming in our later analysis of the model data, and keep in mind that the modelled snowpack density profile likely differs from reality.

### 2.4.1 Snow drift module

The snow drift scheme we used is based on the `snow2blow` model, initially developed by Sauter et al. (2013). Previously, offline simulations with `snow2blow` were forced with WRF input data (e.g., for simulations of blowing snow over the Vestfonna icecap, Svalbard; Sauter et al., 2013), but recently `snow2blow` was directly implemented in the WRF code by Schmid (2021) to allow coupled simulations (i.e., feedback to the atmosphere). While the detailed description of the module and the implementation in WRF is subject to another paper (Saigger et al., 2023), we outline the governing equations and most relevant features of the scheme in the following paragraphs.

The scheme builds on the widely-used approach of dividing the process of drifting snow into a saltation layer and snow particles in suspension, where snow particles are transported by the resolved wind field and turbulent diffusion, as well as being subject to gravity-driven subsidence and sublimation. In the model, suspended snow particles are treated as a passive tracer, so that advection and turbulent diffusion are handled by WRF-internal schemes, while subsidence and sublimation are parameterized. The saltation layer is fully parameterized and acts as a lower boundary condition for the flux of snow into suspension. The drifting snow mainly interacts with the mean flow, while neglecting particle interactions. The mass conservation of snow particles is given by the continuity equation

$$\frac{\partial \phi_s}{\partial t} + \frac{\partial (\phi_s u_i)}{\partial x_i} = \frac{\partial}{\partial x_3}\left(\nu_t \frac{\partial \phi_s}{\partial x_3} - V\phi_s\right) + \left(\frac{\partial \phi_s}{\partial t}\right)_{sub}, \tag{1}$$

where $\phi_s$ is the mass concentration of snow particles in the suspension layer, $\partial \phi_s / \partial t$ is the local rate of snow concentration change, $x_i$ are the Cartesian coordinates, $u_i$ are the Cartesian components of the velocity vector, $\nu_t$ is the turbulent viscosity, and $V$ is the terminal fallout velocity. The fallout velocity

$$V(z) = -\frac{A}{r(z)}\sqrt{\left(\frac{A}{r(z)}\right)^2 + B \cdot r(z)}, \tag{2}$$

depends on the snow particle radius at height $z$

$$r(z) = r_0 \cdot z^{-0.258}, \tag{3}$$

$A$ and $B$ are constants and are calculated with:

$$A = \frac{6.203 \cdot \nu_{air}}{2} \tag{4}$$

and

$$B = \frac{5.516 \cdot \rho_{ice}}{4 \cdot \rho_{air}} \cdot g. \tag{5}$$

Here $\nu_{air}$ represents the viscosity of air, $\rho_{ice}$ the pure ice density and $g$ the acceleration due to gravity. $r_0$ is the particle radius at ground level following (Gordon et al., 2010) with:

$$r_0 = 0.5 \left( \frac{7.8 \cdot 10^{-6} u_\star}{0.036} + 31 \cdot 10^{-6} \right), \tag{6}$$

and $u_\star$ the friction velocity. Optionally, $V(z)$ and $r_0$ can be set to constant values in the model settings.

The last term in Equation 1 accounts for the mass loss of suspended snow due to sublimation based on the formulation of Thorpe and Mason (1966), where the sublimation-loss rate of suspended snow is approximated by $\psi_s \phi_s$, with $\psi_s$ as the sublimation-loss rate coefficient. This coefficient describes the change of snow particle mass due to heat exchange, and ventilation effects. The scheme considers the effect of sublimation on the vertical temperature and humidity profiles in the boundary layer. This feedback mechanism self-limits the sublimation process, because its intensity depends on the saturation deficit of the atmospheric environment (Sauter et al., 2013).

In the saltation layer, snow mass concentration is gained by aerodynamic entrainment from the snowpack below. Snow transport occurs when the surface shear stress exceeds the cohesive bond of the particles. The erosional mass flux is therefore proportional to the excess surface shear stress:

$$q_e = e_{salt} \rho_a \left( \left[ (u_{th} - u_*) \left( \frac{\phi_{salt}}{\phi_{max}} \right)^2 + u_* \right]^2 - u_{th}^2 \right), \tag{7}$$

where $\rho_a$ is the air density, $u_*$ the surface shear stress, $\phi_{salt}$ the concentration in the saltation layer, $u_{th}$ the friction threshold velocity, and $\phi_{max}$ the maximum particle concentration in the saltation layer. Since the particle erosion process depends on the cohesive bonds of the snow particles, the snow density $\rho_s$, which is not affected by snow drift in the model (Walter et al., 2004):

$$u_{th} = 0.0195 + (0.021 \sqrt{\rho_s}). \tag{8}$$

The efficiency of the erosion process is governed by the heuristic parameter $e_{salt}$ [-]. Particle drag reduces the momentum, which in turn limits the capacity to eject further particles. When $\phi_{salt}$ reaches $\phi_{max}$, the friction velocity reduces to the friction threshold velocity and the release of snow particles is stopped. The upper limit of $\phi_{max}$ is given by the semi-empirical relationship (Pomeroy and Male, 1992),

$$\phi_{max} = \frac{\rho_a}{3.29 \cdot u_*} \left( 1 - \frac{u_{th}^2}{u_*^2} \right). \tag{9}$$

When the friction velocity drops below the threshold, particle deposition takes place. The deposition flux $q_d$ corresponds to the downward flux and the modified shear stress ratio

$$q_d = V\phi_s \cdot \max\left(\frac{u_{th}^2 - u_*^2}{u_{th}^2}, 0\right). \tag{10}$$

The first term on the right-hand side describes the vertical turbulent mixing of the snow and the terminal fall velocity $V$, while the second term shows the effect of sublimation in snow mass flux change.

## 3 Results

First, the observed snow depth changes from the TLS acquisitions are introduced and discussed in detail. We explicitly note here that the observations from the TLS data only show the snow depth changes, without distinguishing between different processes. Thus, in the next subsections, we explain the accompanying processes with the aid of further observations at point scale. We use the observations to evaluate the results from the LES, especially in terms of wind patterns and the resulting snow redistribution. Last, we compare the observed and modelled spatial patterns.

### 3.1 Observed snow depth changes

The three TLS scans reveal the changes in snow depth over several days, and also show the heterogeneous snow distribution over the glacier and its surroundings. First, the DEM of Difference (DoD) between scan 1 and 2 shows an increase of surface elevation over almost the entire area of interest (Fig. 3a). The surface elevation increase is snowfall: the precipitation gauges at IHE and StHE registered precipitation, and the snow depth sensor at AWS28 observed a snow depth increase as well (Fig. 4). The snow was evenly distributed over the glacier surface, but the slopes adjacent to HEF showed a more heterogeneous snow distribution between scan 1 and 2 (i.e. around StHE, Fig. 3a), which might indicate preferential snow deposition and/or snow redistribution during the snow fall event (e.g. Mott and Lehning, 2010). From the TLS data, 0.28 m of snow were deposited on average over the glacier and the snow depth sensor observed an increase of 0.45 m between scan 1 and 2 at AWS28. In this study, we do not elaborate on the snow depths at IHE and StHE, as the terrain-dependent snow cover dynamics are unrepresentative at these two stations compared to the rather smooth and homogeneous glacier surface around AWS28.

The DoD between scan 2 and 3 shows a general decrease of the snow depth over the glacier of 0.079 m on average (Fig. 3b). This is in agreement with the snow depth observations at AWS28, where a decrease of 0.08 m was observed by the snow depth sensor between scan 2 and 3 (Fig. 4). A zoom in on the glacier surface on the orographic left side of the glacier (Fig. 3c) and a look at the webcam images reveal patterns which are likely the results of snow redistribution, given their spatial structure. On the glacier surface around AWS28 (pink dot, Fig. 3c), a wavy pattern is evident with magnitudes between approximately -0.15 and -0.05 m. This is comparable to snow bedform observations over similar flat surfaces (Filhol and Sturm, 2015; Kochanski et al., 2018). With the resolution of the snow structure at $\Delta$x=1 m and the webcam images, we cannot distinguish between the snow bedforms (i.e. waves, dunes, barchans or ripples), but the structure is wind-driven. At the slopes adjacent to the glacier

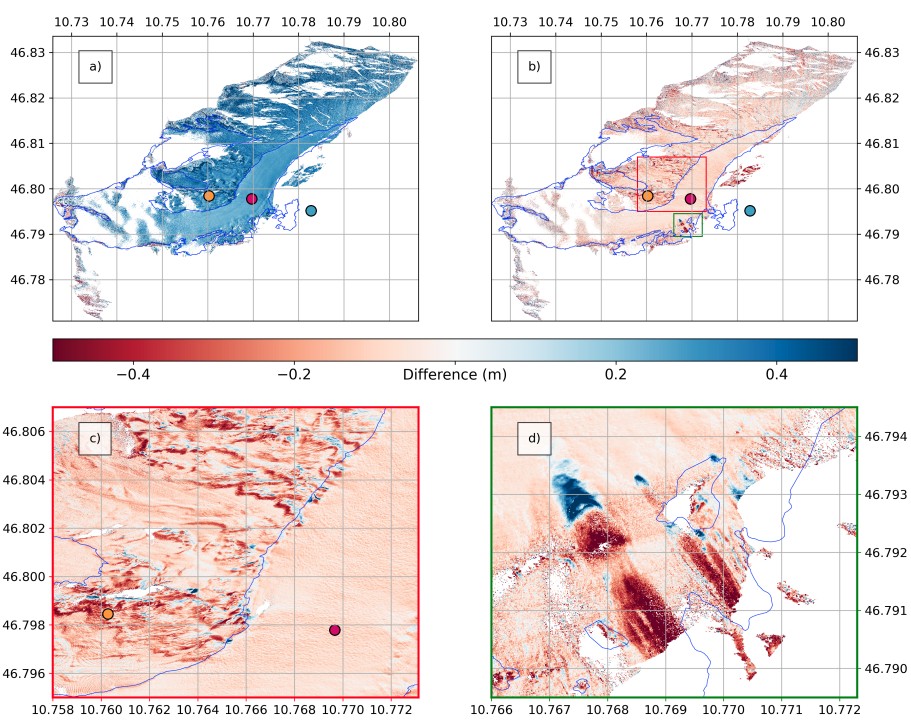

**Figure 3.** DEM of Difference of the TLS scans ($\Delta x=1\,m$) between a) scan 1 and scan 2, and b) scan 2 and scan 3. c) is a zoom of the red box in b) showing signs of snow redistribution, and d) is a zoom of the green box in b) showing avalanches. The glacier outlines (blue) are derived from the ALS data acquired by the Federal Government of Tyrol in 2018. IHE (blue), StHE (orange), and AWS28 (pink) are also plotted.

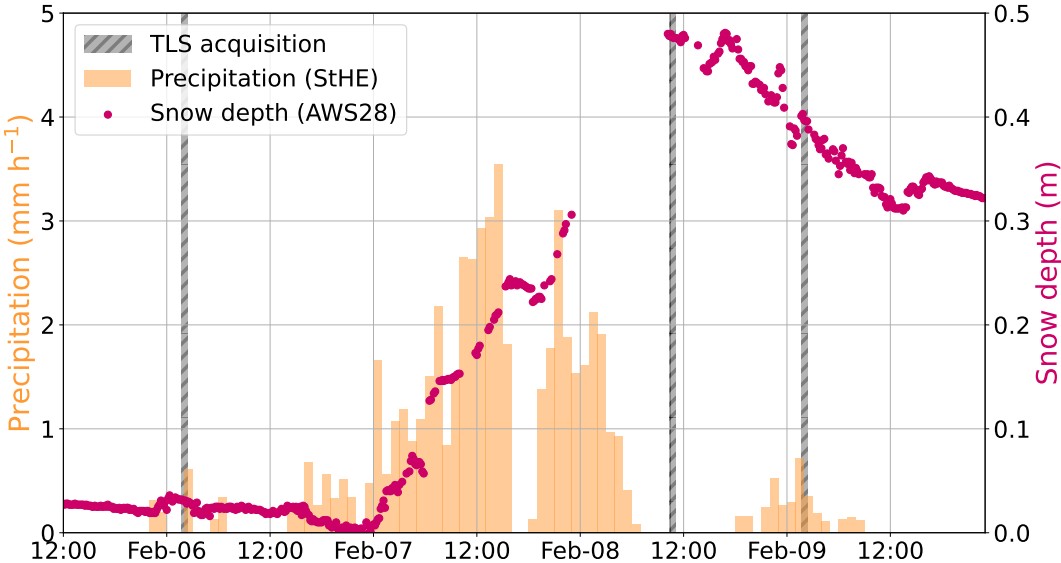

**Figure 4.** Precipitation (StHE: orange) and relative snow depth observations (AWS28: pink) during the case study period. Note that the precipitation observation is not corrected for undercatch and the snow depth is arbitrarily chosen to start at 0 m at the minimum observed during the case study period, despite the glacier was already covered in snow and thus, the actual snow depth was more than 0 m. The grey bars indicate the time of the TLS acquisitions.

surface, snow erosion is observed at the windward southwest slopes and this snow is deposited directly at the closest northeast leeward slopes. This is particularly evident around the location of StHE and the orographic left side of HEF (Fig. 3c). These structures are mainly induced by the rough surface caused by rocks at the slopes surrounding the glacier and again, indicate wind-driven snow depth changes.

Avalanches, induced by fresh snow or wind slabs, are observed over the orographic right side of HEF at an altitude of 2910 m a.s.l. (Fig. 3d). The release zone of the avalanche with magnitudes between -0.25 and -0.68 m is indicated by the dark red color, whereas the dark blue zone shows the deposition area of the avalanche up to +1.26 m high.

We elaborated on the elevation changes at the glacier and the surroundings from the TLS observations, but we aim to investigate the nature of these changes. The surface elevation changes are caused by redistributed snow, compaction, and sublimation. Surface temperature observations from AWS28 below freezing point suggest that melt can be excluded at the glacier during the study period; the simulations also suggest that surface temperatures remain below freezing point over the entire glacier. To distinguish between these processes, we now analyse the additional meteorological observations and the numerical simulations.

| Station | wind speed bias (m s$^{-1}$) | wind speed RMSE (m s$^{-1}$) |
|:---:|:---:|:---:|
| StHE | 2.35 | 3.53 |
| IHE | 2.08 | 4.23 |
| AWS28 | 0.96 | 1.81 |

**Table 2.** Bias and RMSE (15-min data) of wind speed calculated for the three weather stations (StHE, IHE, and AWS28), averaged over 24 hours of simulation time.

## 3.2 Meteorological situation at the glacier: Observations and Simulations

### 3.2.1 Precipitation

Mostly small precipitation amounts were registered on 5 and 6 Feb at StHE (Fig. 4). The situation changed when pre-frontal
precipitation approached our area of interest (7 Feb, around 00:00 UTC). Webcam images, the precipitation gauge, and the
increasing snow depth at AWS28 suggested that fresh snow was accumulated on 7 and 8 Feb. Precipitation stopped at around
at 06:00 UTC on 8 Feb. Precipitation was also registered during the acquisition of scan 1 and 3, but this precipitation was not
evident in the TLS data and on the webcam images. The precipitation during scan 1 was actually registered after the TLS
acquisition, as seen in the 10-min data. Furthermore, we speculate that the precipitation registered during scan 3 is drifting
snow that is captured by the precipitation gauge.

Additionally, precipitation observations are subject to undercatch, which is a well-known problem for precipitation (Goodison et al., 1998; Rasmussen et al., 2012; Smith, 2007; Colli et al., 2014). The data of the precipitation gauge at IHE is not analysed here, as the gauge is placed at a wind-exposed ridge that is hardly ever covered with snow and is thus prone to large amounts of undercatch. The timing and registration of snowfall support the assumption that the snow depth increase between
scan 1 and scan 2 (Fig. 3a) was due to solid precipitation.

### 3.2.2 Wind speed and direction

We now mostly focus on 8 Feb, since on this day the snow drift event of interest occurred. Observed time series of wind speed and direction at the glacier and its surroundings on 8 Feb (Fig. 5) suggest low wind speeds (less than 5 m s$^{-1}$) with mainly northerly flow during the night and the morning hours (8 Feb, 00:00 UTC–09:00 UTC). The wind direction changed
towards southwesterly at around 09:00 UTC, while wind speed increased to more than 5 m s$^{-1}$ at all stations. The wind speed increased even to more than 10 m s$^{-1}$ at the south-facing slope (StHE) and the crest (IHE), while the wind speeds on the glacier remained below 10 m s$^{-1}$ (AWS28). After 15:00 UTC, observed wind speeds increased to more than 10 m s$^{-1}$), which allowed for wind-driven snow distribution.

Similar to the weather station observations, the model simulates low wind speeds during the nighttime at all three stations.
The frontal passage (06:00 UTC-12:00 UTC) is the time period with the highest discrepancy between model and observations. The sudden increase in wind speed sets in an hour earlier than in the observations (Fig. 5), together with an earlier increase in

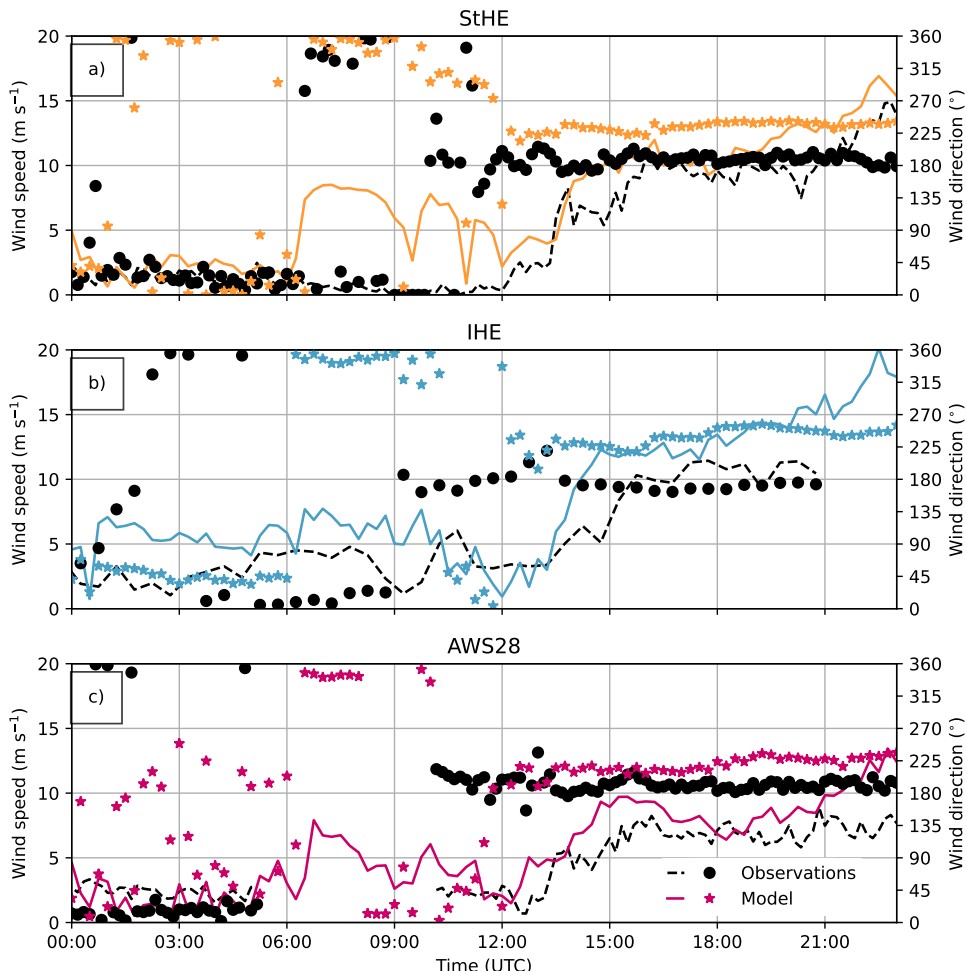

**Figure 5.** Time series of observations (black) from the three weather stations and corresponding model output from the closest grid point in the model (IHE: blue; StHE: orange; AWS28: pink) of wind speed (lines) and wind direction (dots/squares) at 8 Feb 2021. Missing values in the observations are at IHE from 21:00 UTC onward, and at AWS28 between 05:45 UTC and 11:00 UTC.

wind speed, especially at StHE. Furthermore, the wind direction deviated slightly (below 30°) from the observations throughout the simulations. Both the observations and the model suggest dominating southwesterly wind directions with wind speeds over $5\,\mathrm{m\,s^{-1}}$ at all three stations after 8 Feb, 12:00 UTC. This agrees with the TLS observations (Fig. 3c), which also indicate snow redistribution due to these strong, southwesterly winds. We calculated the bias and root-mean square error (RMSE) values over the 24 hours of simulation time after Equations 13 and 14 in Goger et al. (2019) for the horizontal 15-min wind speed of the three stations (Tab. 2). Bias values suggest a wind speed overestimation at all three stations, while this overestimation can likely be attributed to the front passage phase. The best model performance is found for the station on the glacier (AWS28), our primary location of interest.

Overall, judging from the observations we have, the model simulates the wind field on the glacier and the surroundings reasonably well and we assume that the model provides good input conditions for the snow redistribution scheme. However, we have to keep in mind that our set of wind observations is limited and we cannot assess on the correct simulation of the spatial patterns of the wind fields.

### 3.2.3 Compaction, Snow Water Equivalent and snow redistribution

The TLS observations do not give information on the individual contributions of redistributed snow, sublimation, and compaction to surface elevation changes. Compaction of snow can be detected, if the SWE remains constant after a snow fall event together with a simultaneous, continuous decrease in snow depth. SWE observations are not available during the case study period. To investigate the possible amounts of compaction at HEF, we had a look at data from an AWS that was installed at HEF in the winters of 2021/22 and 2022/23 at an altitude 3030 m a.s.l. and provides SWE and snow depth data (Schröder,
2023). In these two winters, we examined snow depth and SWE data of nine snow fall events with amounts between 0.14 m and 0.38 m of fresh snow at the AWS. 16 hours after the snow fall, the snowpack decreased between -6.5% and -25% of the respective fresh snow amounts. In the mean time, no significant changes in the SWE were observed. Even though the winters are not directly comparable (i.e., the winter of of 2021/22 had extremely low precipitation amounts, Voordendag et al., 2023b), the order of magnitude of compaction indicated that this process also likely occurred between scan 2 and 3 in February 2021.
Furthermore, similar amounts of compaction are observed on other glaciers and snowpacks (Gugerli et al., 2019; Koch et al., 2019; Voordendag et al., 2021a). When we apply the compaction rates to the 0.28 m of fresh snow in the case study period, we find that between 0.018 m and 0.071 m of the surface elevation decrease can likely be attributed to compaction.

  We now utilize the model output for further process understanding with a *qualitative* analysis of the modelled snowpack. This allows us to understand possible processes governing snowpack formation; therefore we start with snow redistribution
at point scale. In the model, snow redistribution only occurs when the parameterised friction threshold velocity is exceeded by the current friction velocity, and this value depends on the snow density (Eq. 8). Snow drift and subsequent redistribution is therefore only simulated after the increase of the wind speed to more than $10 \, \text{m s}^{-1}$ after 14:00 UTC. The simulated snow redistribution is found to be -0.022 m at IHE, -0.014 at StHE, and -0.003 m at AWS28 at the end of the simulation period (Fig. 6a). The differences between the weather stations is directly related to the higher wind speeds at IHE and StHE than on
the glacier at AWS28 (Fig. 5). Furthermore, it is interesting to note that the onset of snow drift initially leads to mass loss at all stations, at AWS28, however, snow drift briefly accumulates snow again after 18:00 UTC, while at the end of the simulation, the overall effects of snow drift is mass loss. At the other two stations (StHE and IHE), snow drift continuously contributes to snow mass loss.

  Simulated precipitation and changes in SWE give more insights on mass changes in the snowpack. The simulated pre-frontal
precipitation at StHE agrees well in terms of magnitude and duration with the observed precipitation amounts (Fig. 6b). One of the differences is that the precipitation stops earlier in the model than in the observations. We conclude that the model is able to simulate the temporal pattern on the case study day successfully, albeit with a slight underestimation. During the pre-frontal

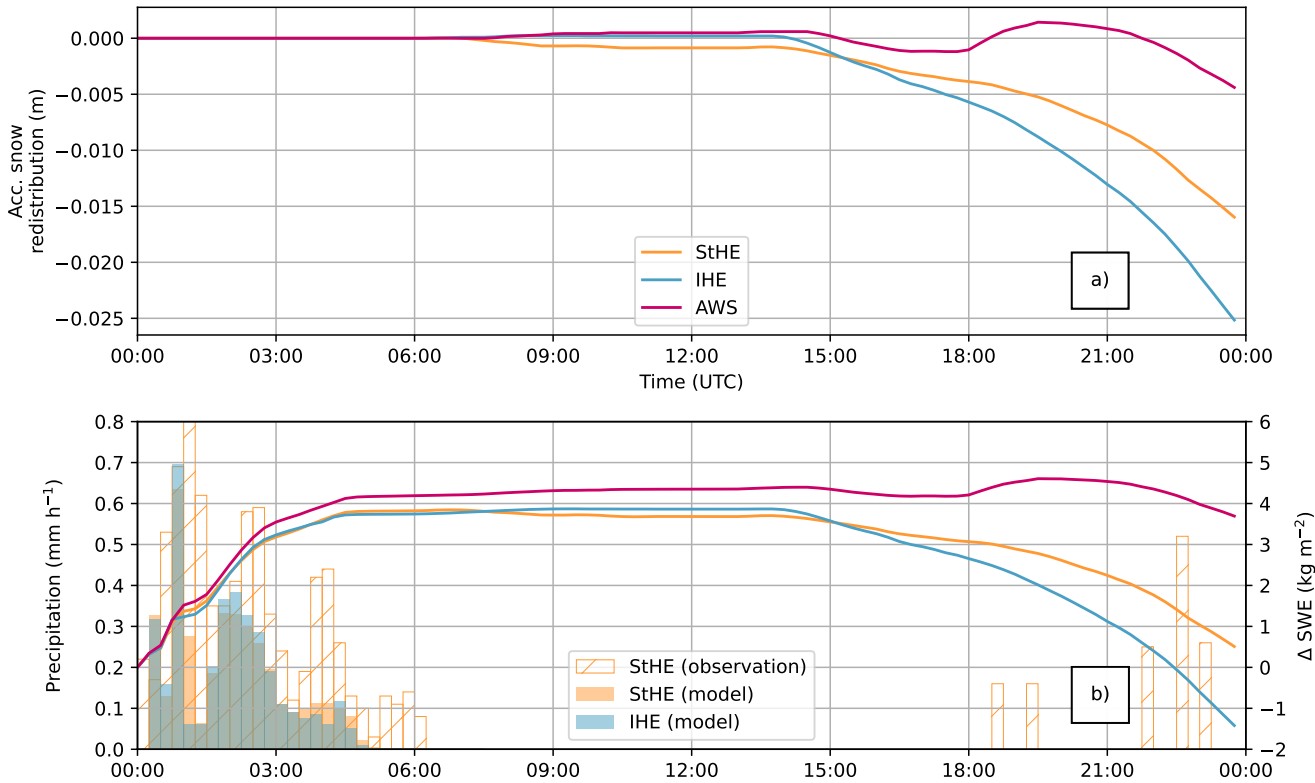

**Figure 6.** Time series at 8 Feb 2021 from the closest grid point in the model to StHE (orange), IHE (blue), and AWS28 (pink) in panel a) of accumulated snow redistribution (relative to the start of the simulation at 00:00 UTC), and panel b) solid precipitation (bars), and the relative change of snow water equivalent ($\Delta$ SWE, lines).

precipitation period, the simulated SWE also increases, with similar values for IHE and StHE, but with about $0.4\,\mathrm{kg\,m^{-2}}$ higher values of SWE at AWS28 (Fig. 6b).

After snowfall ended and before snow drift started (05:00–15:00 UTC), the simulated SWE values remain constant for the three stations, indicating that the surface elevation change during this period (Fig. 4) is snow compaction. As soon as snow drift started, SWE reduces as snow gets eroded at the locations of the stations, but with spatial differences between IHE, StHE and AWS28 (Fig 6). The ridge location IHE exhibits the largest reduction in SWE due to its exposed location over the entire simulation time. AWS28, however, shows a smaller total loss of SWE, mainly because of the sheltered location of the glacier.

To summarize, SWE increased until 06:00 UTC due to solid precipitation, while SWE remained constant until 15:00 UTC as only compaction took place. After 15:00 UTC snow redistribution led to a continuous decrease in SWE at all stations. Thus, the model suggests that snow mass changes due to snow redistribution do not occur until 15:00 UTC. The increase in horizontal wind speed after 12:00 UTC triggers snow drift in the model. At AWS28 snow erosion reduces the SWE after 15:00 UTC and deposition takes place after 18:00 UTC. At the other two locations (StHE, IHE), SWE is constantly reduced by snow erosion.

Higher wind speeds result in more snow particles in the air and a higher likeliness of sublimation. Along with that, sublimation also depends on the vapor pressure in the ambient air as well as on the snow particle size. However, the values of simulated sublimation remain very small (less than 1 kg at the end of simulation for the entire air column over all glaciated areas in Fig. 1) throughout the rest of the simulation (not shown). Therefore, we will not discuss this process in more detail, also because the simulated sublimation contribution to snow mass loss is much smaller than the uncertainty of the TLS.

### 3.3 Spatial patterns of simulated snow redistribution processes

To explore the spatial patterns beyond the point scale, we analyse the simulated snow redistribution relative to the start of the simulation at 8 Feb, 00:00 UTC on the glacier and its surroundings (Fig. 7). The simulated snow redistribution is given in meters, to be able to compare it to the surface elevation data of the TLS. The simulated wind arrows reveal wind speeds over $5\,\mathrm{m\,s^{-1}}$ during the wind-driven snow redistribution phase, and the corresponding wind direction was mostly down-glacier (South-Westerly), in agreement with the observations from AWS28 (Fig. 5). In the model, the governing process for snow
redistribution is erosion (Fig. 7a-d), which was especially strong at the mountain ridge northwest of HEF. This is in accordance with the webcam images, which suggested that snow erosion mainly occurred at the surrounding mountain ridges. Snow deposition (Fig. 7e-h), however, was very small compared to erosion and does not exceed 0.01 m after 24 hours of simulation time. The only exception is the short phase at the glacier at AWS28, where a small increase in snow depth is noticeable around
18:00 UTC in both observations (Fig. 4) and simulations (Fig. 6).

   Therefore, the final snow redistribution (Fig. 7i-l) in the model is mainly governed by erosion, but some areas on leeward slopes experience more deposition than erosion (Fig. 7l, e.g., around coordinates 46.795°N, 10.82°W). At the end of the simulation, the model suggests that around 0.09 m of snow were eroded at the mountain ridges, while at the glacier 0.03 m of snow were eroded. Although there was a weak positive signal in snow redistribution in the vicinity of AWS28 at 15:00 UTC
(Fig. 6 and 7i), the sum of erosion and deposition resulted in an overall decrease in snow cover on the glacier. A main reason for this were the high wind speeds throughout the domain; wind speeds reach up to more than $10\,\mathrm{m\,s^{-1}}$ after 15:00 UTC. Therefore, we conclude that snow can be easily eroded and transported towards the North-East and out of the domain.

### 3.4 Direct comparison of simulated and observed snowpack changes

The snow depth changes from the observations (between 8 Feb, 10:22 UTC and 9 Feb, 01:42 UTC, Fig. 8a) are compared to
the simulated snow redistribution (between 8 Feb, 10:15 UTC and 9 Feb, 00:00 UTC, Fig. 8b). A similar snow redistribution pattern as in the model also appears in the snow depth change observations by the TLS calculated to the model grid size of $\Delta x$=48 m between scan 2 and 3 (Fig. 8a,b). In the simulation, most snow is redistributed away from the mountain ridges, as we also observed in the webcam images (Fig. 2) and during fieldwork campaigns. Nevertheless, the area in the accumulation zone of the glacier and at the ridges is sparsely covered by the TLS, but we observe in both the observation and the simulations that
the snow is evenly distributed over the glacier tongue (e.g., around AWS28). However, when the high-resolution TLS data of Fig. 3 is upscaled to $\Delta x$=48 m, many of the detailed structures (Fig. 3c) disappear at the model's resolution. The spatial patterns of simulated SWE (Fig. 8c) suggest a close connection to the snow redistribution patterns in (Fig. 8c); the general decrease in

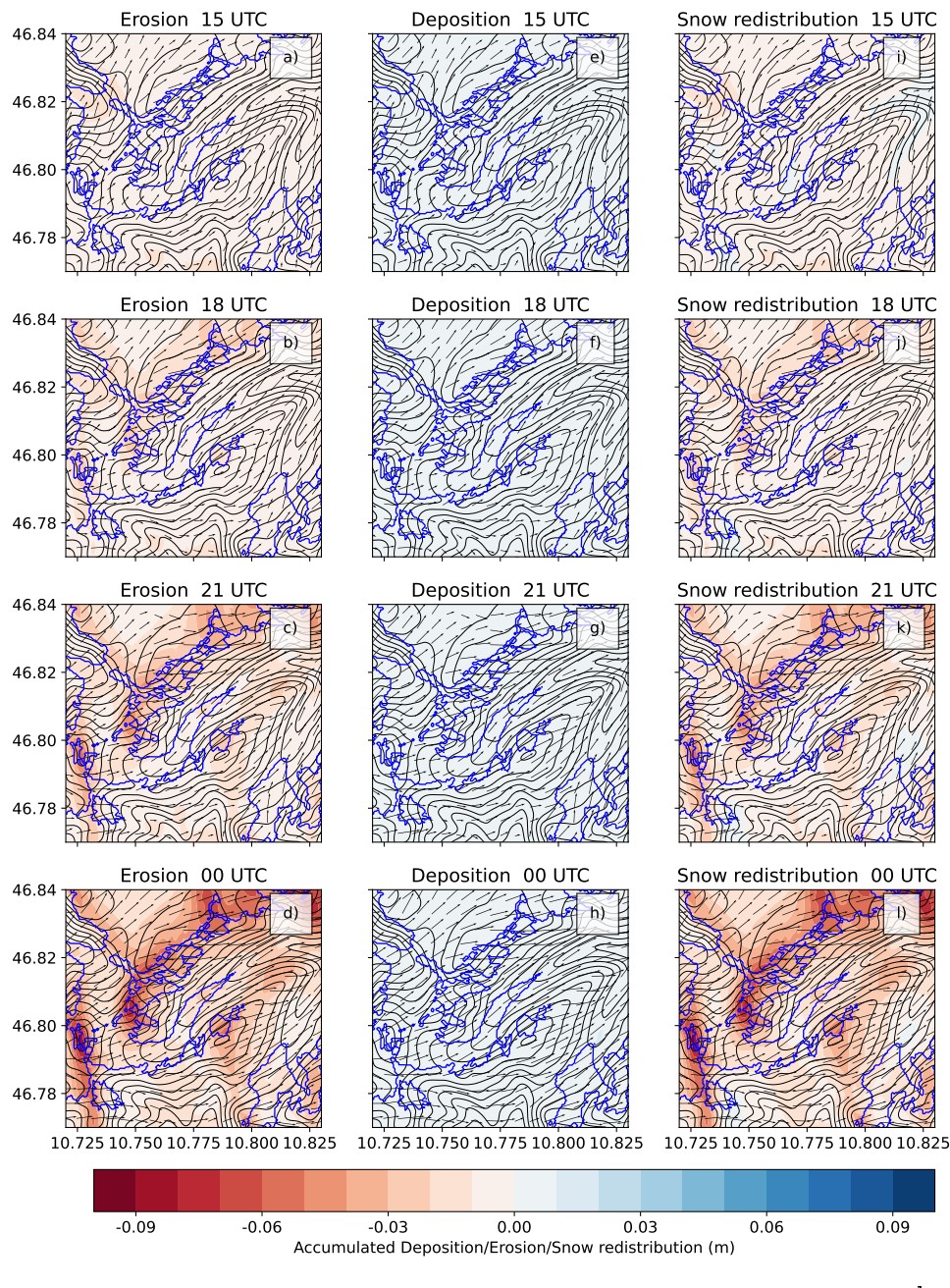

**Figure 7.** Simulated snow erosion (a)-d)), snow deposition (e)-h)) and the resulting net snow redistribution (i)-l)) in colors from 15:00 UTC, 18:00 UTC, 21:00 UTC and 00:00 UTC. Note that all values related to snow redistribution are summed up from simulation start (8 Feb, 00:00 UTC). Horizontal near surface wind speed and direction are indicated by black arrows. The black contours of 100 m equidistance show the model topography, blue contours indicate the glacier outlines.

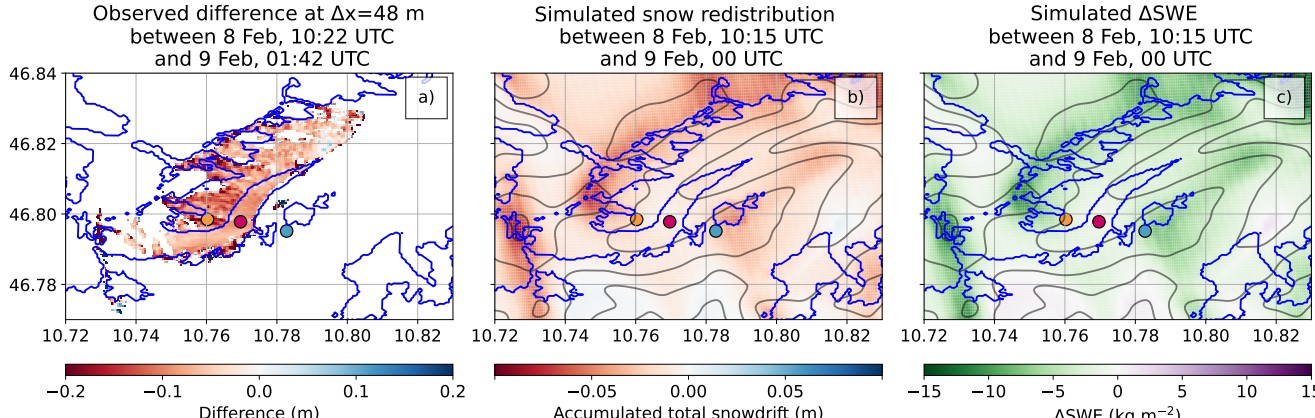

**Figure 8.** a) Observed snow depth changes over HEF at $\Delta$x=48 m between 8 Feb, 10:22 UTC and 9 Feb, 01:42 UTC, and b) the simulated snow redistribution and c) the simulated change in SWE between 8 Feb, 10:15 UTC and 9 Feb, 00:00 UTC. Note the different orders of magnitude in a) and b). The glacier outlines as used by the model are given in blue.

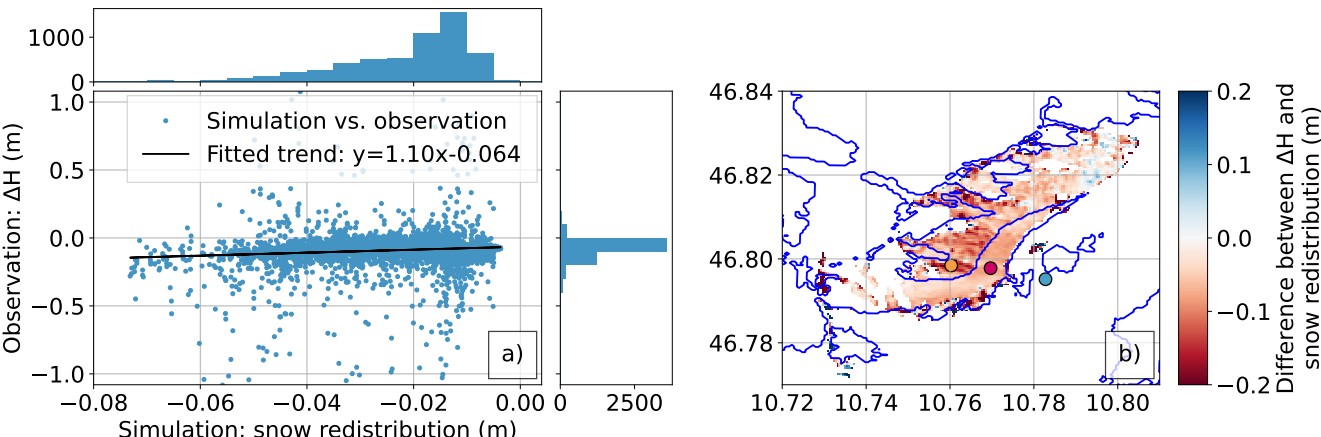

**Figure 9.** a) Observed snow depth changes over HEF at $\Delta$x=48 m between 8 Feb, 10:22 UTC and 9 Feb, 01:42 UTC plotted against the simulated snow redistribution for all the covered grid cells (blue) and the linear fit between these variables (black). The subplots show the density distribution of observed snow depth changes and simulated snow redistribution, respectively. b) The difference between the observed snow depth changes and the simulated snow redistribution over the region of interest. The glacier outlines as used by the model are given in blue.

SWE directly corresponds to the snow erosion patterns at the mountain ridges. The average simulated decrease caused by snow redistribution is -0.026 m over the glacier (Fig. 8b), which equals a decrease in SWE of -3.9 kg m$^{-2}$ (Fig. 8c) in the simulated period over HEF. The order of magnitude of the snow depth changes from the observations is twice as large as the simulated snow redistribution due to snow drift from the simulation. The observations from the TLS do not give information whether


the snow depth changes occur due to snow drift or compaction. In Sect. 3.2.3, we found that between 0.018 m and 0.071 m of the surface elevation decrease can be attributed to compaction. We quantify the amount of compaction during our case study by comparing the observed surface elevation changes and the simulated snow redistribution for each grid cell covered by the TLS system (Fig. 9a). After fitting a linear trend through these data, a relation can be detected from the simulations $S$ and observations $O$:

$$O = 1.10S - 0.064 \tag{11}$$

The relation suggests that for every 0.01 m of simulated snow distribution 0.011 m of snow redistribution is observed, or in other words, the model underestimates the amount of snow redistribution by only 9.1%. We assume that the compaction rate over the snowpack in the period of 15 hours over the study area is constant and thus the 0.064 m in Equation 11 is related to the compaction of the snowpack. This amount of compaction is in the range of the compaction that we found for a different winter season (between 0.018 m and 0.071 m of the total snowpack decrease of 0.079 m). Therefore, we assume that the average compaction rate of 0.064 m over 15 hours during this study period is realistic. The distribution density (Fig. 9a) of the observational data is more variable compared to the corresponding model results, suggesting that the model is not able to capture the full complexity of wind-driven snow redistribution. This is related to the more complex real topography compared to the smoother model topography. Furthermore, events such as avalanches are not represented in the model. Likewise, the amount of compaction is not absolutely constant over the study area, as this also depends on the snow depth and the weight of overburden layers, and to a minor extent to the wind speeds. However, we assume that variability in compaction is low relative to the effects of snow drift and therefore assume it to be constant.

The observed snow redistribution amounts are subtracted from the observed surface elevation changes in Fig. 9b, which theoretically gives information on a model bias and realism of the spatial pattern of snow redistribution. However, we have to keep in mind that the TLS data includes the snowpack compaction, and the amount of the observed snow redistribution is small. Adding the spatial average of the snow compaction rate from Fig. 9a to the observational data set leads to inconsistencies; therefore, we omit this step. However, the spatial patterns of snow depth change suggest that the model is not able to capture the small-scale snow depth structure at the slopes. Yet, at the rather "flat" glacier surface (compared to the surroundings), the spatial structure of the simulated model patterns agrees well with the TLS observations. This result is relevant for the question whether the snow drift module can be used for further glacier mass balance research.

## 4 Discussion

The present study combines operational TLS observations and LES for a case study to detect snow redistribution on an Alpine glacier. Since this is a small-scale phenomenon, it pushes both observations and modelling towards their boundaries.

The observations with the permanent TLS station are worldwide unique. Other studies also investigated snow depth changes with TLS (e.g. Mendoza et al., 2020; Gabbud et al., 2015; Fey et al., 2019) or ALS (Helfricht et al., 2014; Grünewald and Lehning, 2011), but these studies mainly covered coarser temporal resolutions or only covered small parts of a glacier. We

were able to capture snow fall and redistribution directly thereafter, but we also note that the TLS observations are at the limits

of the capabilities of the system. The uncertainty of the TLS observations was estimated at $\pm 0.10$ m in vertical direction with manual post-processing in Voordendag et al. (2023a). However, the registration is even better in vertical direction if we look at the registration of the scans at the manually selected tie objects in this study. Thus, the snow depth changes between scan 2 and 3 were measured reliably and in agreement with observations from the snow depth sensor at the glacier.

Still, a systematic evaluation of snow transport models with observations is challenging. In our case, the pixel-to-pixel

comparison between the model and the TLS observations allowed us first insight on model performance, however, we are aware that we are comparing different terrain geometries between model and observations. On the other hand, point observations of snow depth or blowing snow fluxes might be unrepresentative, because spatial variability is especially high in complex terrain. New observational approaches such as particle tracking velocimetry (Aksamit and Pomeroy, 2016) will allow for more detailed evaluation of high-resolution snow transport models. Furthermore, bringing modern, multi-scale observational

methods together (e.g., TLS, particle tracking velocimetry, snow depth and SWE measurements) in dedicated measurement campaigns would provide excellent test beds for snow model validation.

Modelling small-scale boundary-layer processes over mountainous topography is still a challenge for a NWP model like WRF, as discussed in the previous summer study by Goger et al. (2022). However, compared to the summer study, the model simulated even more realistic wind patterns over the glacier and its surroundings. Therefore, we assume that no model bias

emerges due to erratic wind patterns. Still, we have to keep in mind, that these promising simulation results only apply to our case study and can be different for other time periods or locations. The simulated snow redistribution is realistic in terms of spatial structure. However, the processes at smaller scales are smoothed out, which is due to the horizontal resolution of 48 m and the smoothed model topography restricted by numerical stability. The model topography limits the slope angles to a maximum of $35°$, and thus the model topography clearly deviates from real topography. In agreement with the TLS

acquisitions, the simulations show that snow is eroded mostly at the ridges and that the snowpack at the glacier is sheltered and less affected by snow erosion.

High wind speeds immediately redistribute freshly deposited snow again, until it is transported out of the domain, therefore, erosion strongly dominates. Also, the very small-scale snow redistribution areas (Fig. 3c) cannot be captured at a $\Delta x$=48 m, since Mott and Lehning (2010) noted that $\Delta x$=10 m or less would necessary to calculate the small-scale deposition patterns we

observed with the TLS on the glacier. Still, we assume that the general snow redistribution patterns are well-simulated, as the model captures the larger snow redistribution at the mountain ridges and smaller snow redistribution and lower wind speeds at the less exposed parts of the glacier in agreements with weather station and TLS observations.

One of the advantages of the presented snow drift module in WRF is its simplicity compared to fully coupled atmospheric and snow models (Vionnet et al., 2013; Sharma et al., 2023), because our snow drift scheme are embedded within the estab-

lished modules of the WRF modelling system. However, coupling to grain-scale snow models (Vionnet et al., 2013; Sharma et al., 2023) can, of course, provide more detailed information on snowpack evolution and full feedback (fluxes, temperature, humidity) between the atmosphere and the snowpack is possible. In our setup, the feedback of the atmosphere by the snow drift module consists of the impact of snow sublimation on the temperature and special humidity of the atmosphere aloft (Saigger

et al., 2023). Furthermore, employing a full physics-based atmospheric model at high resolution provides high-resolution input data for the land surface model. This poses an advantage compared to completely uncoupled hydrological systems (e.g., Marsh et al., 2020; Quéno et al., 2023; Baron et al., 2023), which rely on input from downscaled data, which can be also challenging over complex topography. The snow drift module is coupled to the WRF code and the land-surface scheme NOAH-MP Nevertheless, NOAH-MP provides only three layers in the snowpack, whereas physical multi-layer snow models, such as SNOWPACK (Lehning et al., 1999), are able to simulate more layers and include a more realistic representation of physical snowpack processes. However, with the aim to investigate the contribution of snow redistribution, it is only necessary to calculate the surface shear stress $u_{th}$ (Eq. 8) depending on the snow density of the upper layer in our snow drift module. The initialisation of the snowpack in our simulation is simplified, as the inner domain of the model is initialised with fresh snow only, because the computationally expensive LES cannot be run with a long spin-up time for snowpack initialisation. Thus, the model lacks accurate information on the long-term snowpack evolution. In nature, the lower layers of the snow are compressed, but the upper layer with fresh snow is still uncompressed. It is more likely that snow drift takes place on an uncompressed, fresh snowpack rather than on a dense snowpack. We consider the snow initialisation in the model unproblematic for this case study, as in both nature and simulation only the fresh snow is eroded. In the model, snow compaction is calculated following Anderson (1976). The results of this snow compaction (not shown) are overestimated, because the model is initialized with a snowpack entirely consisting of fresh snow (>2 m of fresh snow), enabling high compaction rates, whereas in nature there is only the 0.48 m of fresh snow on top of older snow layers available for compaction. Also, the amount of snow at the glacier can be derived with DEM differencing of TLS scans between October 2020 and February 2021, but any of the physical properties of the snowpack, such as surface temperature or density remain illusive, which makes a realistic initialization also not viable. However, we found realistic amounts of wind-driven snow redistribution in our simulations and we therefore conclude that a three-layer model for the snowpack is sufficient to qualitatively assess wind-driven snow redistribution.

Wind-driven snow redistribution contributes to the glacier mass balance (Dadic et al., 2010) and for this specific case study, snow redistribution has a negative effect on the glacier mass balance of HEF. In the simulation -3.9 kg m$^{-2}$ of snow is blown away from the glacier and out of the domain during the simulation period. We only focused on one case study, as the time period was characterized by low wind speeds during snow fall and higher wind speeds with snow redistribution afterwards. Furthermore, AWS28 was installed at the glacier and the second scan was taken directly after snow fall. It is clear that we cannot attribute for the seasonal contribution of snow redistribution for the glacier mass balance with this one "golden day". Further research is needed to investigate this seasonal contribution using our extensive TLS data set, preferably also to investigate snow redistribution patterns under different prevalent wind directions (e.g., Southerly or North-Westerly). Our study shows that a fresh snow fall event and a rapid increase in wind speeds directly thereafter are favorable conditions for snow drift to occur; therefore, snow drift is likely to be present mostly in connection with frontal passages or downslope windstorms.

Finally, although the installation of a permanent TLS station in remote mountainous terrain is a logistical challenge, the WRF model setup could be applied to any location worldwide. Therefore, our model setup can also be utilized for snow redistribution studies at other glaciated areas. In our current set-up, the horizontal resolution is rather high due to the highly complex terrain

of our area of interest ($\Delta x = 48$ m). Still, our set-up can also be applied with coarser grid spacing over large ice sheets over Greenland or Antarctica for seasonal runs.

 ## 5 Conclusions

In this study, we introduced unique TLS scans to validate large-eddy simulations with the WRF model for quantifying the effect of snow redistribution over Hintereisferner, a major Alpine glacier in the Austrian Alps. For this purpose, we present a case study between 6 and 9 Feb, 2021, where multiple TLS scans and additional observations of wind speeds and snow depth on the glacier are available. Webcam imagery revealed snow drift in the area. With this rich observational data set, we evaluated large-eddy simulations at $\Delta x$=48 m with the WRF model including a newly implemented snowdrift module. Our major findings are summarized as follows:

– Surface elevation changes due to snow fall and snow redistribution are observed with three TLS scans between 6 Feb, 01:42 UTC and 9 Feb, 01:42 UTC, 2021. Simulations were performed for 8 Feb, and run for 24 hours. The combination of high-resolution observations and simulations at HEF is able to capture the glacier-wide snow redistribution patterns.

– The TLS scans can deliver information on typical snow redistribution patterns. They show spatial heterogeneity, while on the glacier the patterns are less prominent than on the orographic left slope.

– Observations with the TLS show a glacier-wide spatially averaged decrease of 0.079 m of the snowpack in the 15 hours directly after the snow fall. This reduction of the snow depth is a combination of snow compaction and snow redistribution.

– The large-eddy simulations with the WRF model at $\Delta x$=48 m simulated the wind patterns at the glacier exceptionally well, and a newly implemented snow drift module allows a detailed comparison with the TLS acquisitions. The simulated integrated glacier-wide snow redistribution is on spatial average 0.026 m. The snow redistribution patterns are captured in a realistic manner compared to the observations.

– A qualitative inspection of the simulation results reveals that snow is mostly eroded on the surrounding mountain ridges, while the glacier itself is in a sheltered location and experiences less snow redistribution. The model is able to simulate snow redistribution in a reasonable way, given that the model topography is still smoothed at $\Delta x$=48 m, therefore simulated snow redistribution is smoother than in nature.

– We can estimate the mean snow compaction over 15 hours from the observed surface elevation changes and the simulated snow redistribution during this case study with linear regression analysis. Averaged snow compaction is found to be 0.064 m, and the model underestimates snow redistribution by 9.1%.

– Snow redistribution has a negative effect on the glacier mass balance in this case study with a simulated mass decrease of -3.9 kg m$^{-2}$ in 24 h. However, the contribution of these snow amounts to the seasonal glacier mass balance remains illusive as this study only covers one case study with a specific wind pattern, but this is subject to further research.

- The operational high-resolution observations of surface elevation changes at HEF with the permanent TLS are currently worldwide unique. To obtain similar data sets at other glaciers, similar measurement systems would have to be installed there.

- The WRF model setup with the snow drift module produces reasonable results and can be applied to any other location in the world, when high-resolution static and meteorological input data are available for the location of interest.

This study investigated the impact of snow distribution over a major Alpine glacier. Snow redistribution patterns depend on the wind field and the local topography; therefore, our work shows the potential impact of small-scale boundary layer processes on glaciers' mass balance. Further case studies at HEF, but also at other mountain glaciers would shed more light on the impact of wind-driven snow distribution on glaciers' mass balance. Furthermore, more detailed information of the wind fields and the snowpack will benefit distributed glacier mass balance models such as COSIPY (Sauter et al., 2020).

*Code and data availability.* The snow drift module for WRF can be found in the following Github repository of M. Saigger: https://github.com/manuelsaigger/WRFsnowdrift. TLS data is available upon request from ACINN/R. Prinz, meteorological data can be downloaded from https://acinn-data.uibk.ac.at/pages/station-list.html, and simulation output is available upon request from B. Goger.

*Author contributions.* AV selected the case study period, and conducted and post-processed the TLS observations. BG conducted the WRF simulations and analyzed the model output. AV and BG wrote the original manuscript. RP oversaw the meteorological and mass balance observations at HEF. TS developed the snow drift module and implemented it into the model code, while the code is currently maintained by MS. GK, TM, and TS conceived the project idea and oversaw the entire progress of the project. All authors contributed to the manuscript and improved it where necessary.

*Competing interests.* Tobias Sauter and Thomas Mölg are members of the editorial board of TC.

*Acknowledgements.* This work is part of the project "Measuring and modeling snow-cover dynamics at high resolution for improving distributed mass balance research on mountain glaciers", a joint project fully funded by the Austrian Science Foundation (FWF; project number I 3841-N32) and the Deutsche Forschungsgemeinschaft (DFG; project number SA 2339/7-1). The computational results presented have been achieved using the Vienna Scientific Cluster (VSC) under project number 71434. Christina Schmid is acknowledged for the initial implementation of the snow drift module in WRF. We would like to thank Wolfgang Gurgiser and Philipp Vettori for their assistance in installing weather stations on and around HEF. Christian Georges, Christoph Klug and Rudolf Sailer facilitated the TLS setup. We thank Nora Helbig for editing our article and the two referees for their thoughtful comments leading to a substantial improvement of the manuscript.

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
