# Peer review of "A novel framework to investigate wind-driven snow redistribution over an Alpine Glacier: Combination of high-resolution Terrestrial Laser Scans and Large-eddy simulations"

_EGUsphere, 2023_

## Author Comment (AC1)

**Response to Referees**
**Investigating wind-driven Snow Redistribution Processes over an Alpine Glacier with high-resolution Terrestrial Laser Scans and Large-eddy Simulations**

Annelies Voordendag, Brigitta Goger, Rainer Prinz,
Tobias Sauter, Thomas Mölg, Manuel Saigger and Georg Kaser

October 16, 2023

Dear editor and referees,

We would like to thank the editor for handling our manuscript and the referees for their careful evaluation of our work and the valuable suggestions, comments and questions. We believe that the manuscript will substantially benefit from the referees' feedback. Below we address our detailed responses to all the comments.

In this response-to-review document we try to clarify and address each of the suggestions, comments and questions made during the review. Therefore we have copied the comments in blue boxes and have addressed them one by one. In the response we use italic fonts to quote text from the revised manuscript.

Yours sincerely, Annelies Voordendag, Brigitta Goger & co-authors

**Response to referee #1**

**Overview**

> R1-1: The manuscript investigates the snow redistribution processes over a well monitored glacier in Austria using a unique observational dataset and models. This topic is right now a major hotspot in snow research since snow drift and redistribution has received less attention due to the difficulty of measure and modeling drifting snow. I think that is therefore right to give attention to this topic and increase the efforts on research, as the authors does in this paper through both observations and modeling.

We are very thankful for the constructive comments of referee #1.

> R1-2: The dataset consists in Terrestrial Laser Scans and automatic weather stations, which has been previously exploded by the authors for study the snow dynamics on the site. The main innovation of this research is using a new snow drift module included in WRF to simulate at high resolution in complex terrain. This module adds to other initiatives in the same direction such as the Meso-NH/Crocus (Vionnet et al. 2014) and CRYOWRF (Sharma et al. 2023) without explicating adding a snowpack model as a surface model, and using the broadly used NOAH-MP. However, the fact that the description of the model is not the main focus of this manuscript and that the manuscript of the model is still in preparation, makes difficult to evaluate more deeply the manuscript.

We agree that it was difficult to review the manuscript without the manuscript of the model description. However, the manuscript of this paper has now been submitted and the preprint is available (Saigger et al., 2023). The most essential parts are also explained in our manuscript.

> R1-3: In any case, the conjunction of this observational dataset and the modeling results show promising results for investigating the snow drift. However, I think that considering that the study is based in a unique case study, the analysis of the data can be more deeply considered. My current feeling is that the authors show disentangled the results from observations and from modeling, but they did a small effort on putting together the data and comparing them more quantitatively instead of the current qualitative comparison. Considering

the potential of both the dataset and the numerical approach, I encourage the authors to do an extra effort to evaluate the model during this case study and give extra insights for this interesting case study.

Therefore, I think that this interesting manuscript can be published after addressing my previous concerns.

The TLS data show snow depth changes and these changes are in winter mainly compromised of snow drift and compaction. Direct measurements of snow drift and compaction were not available during the case study period, and therefore we estimated likely compaction rates with data at HEF from two other winter seasons.

The direct comparison of TLS observations and model output is a challenge. However, the referee is right in pointing out that a deeper analysis can be done with our available data. For the revised manuscript we will analyze the observations and simulations further. We plotted the observed surface elevation changes and the simulated snow redistribution for each grid cell covered by the TLS system. After plotting a linear trend through these data (Fig. R1), a relation has been found between the simulations and observations:

$$OBS = 1.1SIM - 0.064 \qquad (1)$$

The relation suggests that for every $1\,\mathrm{cm}$ of simulated snow distribution $1.1\,\mathrm{cm}$ of snow redistribution is observed, or in other words, the model underestimates the amount of snow redistribution by only 9.1%. A bias of $0.064\,\mathrm{m}$ can be related to compaction of the snow pack. This corresponds very well with the compaction rates we found with the SWE and snow depth data at HEF in the other winter seasons where we found that between $0.018\,\mathrm{m}$ and $0.071\,\mathrm{m}$ of the surface elevation decrease can be attributed to compaction. The observed data seems noisy in Fig. R1, but this can be related to the more complex topography in reality than in the model and events such as avalanches.

Additionally, the method to upscale the high resolution plot ($\Delta$x=1\,m) in Fig. 4 to the coarser resolution ($\Delta$x=48\,m) in Fig. 8a has been examined while preparing the comparison between observations and simulations in Fig. R1. Comparing Fig. 4 with Fig. 8a also shows an increase in glacier coverage by the TLS, mainly in the glacier's accumulation zone at the coarser resolution, but this is caused by the method to calculate the $48\,\mathrm{m}$ grids. Within an area of 48 times $48\,\mathrm{m}$ the average of all the returned laser pulses within this area is taken. Often, this is less than 1 point $\mathrm{m}^{-2}$, which is not representative for the entire area. Also, at the edges of the entire scanned surface, problems occurred due to sparse coverage. Therefor, all $48\,\mathrm{m}$ grid cells that are not neighboured by 8 other grid cells or which have less than $48^2$ (=2304) returned laser points are removed from the data set. This led to an adjustment of Fig. 8a.

We will add Fig. R1 and an updated version of Fig. 8a to the manuscript and describe the relation between the observations and the simulations.

**Piecemeal**

R1-4: **Title:** Correct.
**Text:** The text is well and carefully written. Can be easily understood and the language is correct.
**Figures:** Figures are in general correct and labels and references are informative. As commented below, I think that modeling data should be compared with observations in Figure 6a, and that Figure 8 should compare the model and the observations from a more fair perspective (see comments on results)

We agree that a more fair comparison of model and observations is necessary. We referred to the figure in R1-3 and will add it and its description to the manuscript.

R1-5: **Introduction:** Introduction is well written. The background of the study is extensively exposed with significant citations. However, I have some comments on this section:

L33-34: Why only four processes? I think that snowpack is affected by a number of processes including, furthermore those stated by the authors, the creation of a hoar layer, the metamorphosis processes into the snowpack, or the redistribution by gravity on avalanches. Indeed, the processes depends on the scale. I think the authors are referring to the processes that modify the snowpack height only, but not in general.

We agree that more than four processes can affect the change in snowpack. As the referee noticed, we here only refer to the snow depth changes. We propose to reformulate the respective text as follows:
*Furthermore, during or after snowfall, the depth of the snowpack is affected by four processes: melt, compaction, sublimation, and wind-driven snow redistribution.*

R1-6: L71-72: Why, according the authors, coupling a standalone snowpack model to an atmospheric model

[Figure]

Figure R1: Observed surface elevation changes plotted against the simulated snow redistribution for all grid cells with more than 2304 returned laser pulses (red dots). The fitted linear trend between the observations and simulations is plotted (black) and given in the legend.

> is a disadvantage? Indeed, this sentence refers to CRYOWRF, which included a specially developed scheme for drifting snow in WRF such as the authors uses in their research. However, instead of using the three-layer simplified model from Noah-MP as a land surface scheme, they use a more advanced snowpack model, that provides a better representation of the snow, impacting positively to the blowing snow representation. In my opinion, authors should reformulate this statement.

We agree. The sentence will be rewritten as follows:

*To our current knowledge, no treatment of snow-driven redistribution is directly implemented in the land-surface scheme of WRF, NOAH-MP (Niu et al., 2011). The advantage of a snowdrift module directly implemented in WRF would be that no extra compilation with additional code is necessary, and that computational costs can be constrained.*

> R1-7: **Methods:** Methods are correct to describe how the case study was investigated. Authors describe both the observations and the model. While, as the authors stated, the comprehensive description of the module is subject to another paper that is not published yet, main equations used in the blowing snow module are stated. However, since that paper is not published yet, the authors should include some more details, such as the equations used for the sublimation process. In the same way, some basic notions of the numerical implementation of the blowing snow scheme in WRF should be mentioned to a reader may understand how the model computes the blowing snow before the modeling paper is released. This is useful to better understand and evaluate the following sections until the description paper is ready.

The paper on the model setup is now available as preprint: `https://doi.org/10.3929/ethz-b-000636467`. We added more context in the introduction of Sect. 2.4.1:

*The scheme builds on the widely-used approach of dividing the process of drifting snow into a saltation layer and snow particles in suspension, where snow particles are transported by the resolved wind field and turbulent diffusion, as well as being subject to gravity-driven subsidence and sublimation. In the model, suspended snow particles are treated as a passive tracer, so that advection and turbulent diffusion are handled by WRF-internal schemes, while subsidence and sublimation are parameterized.*

[Figure]

Figure R2: Time series of air temperature and wind direction for the three stations and model output.

We also added the reference of Thorpe and Mason (1966) to the sublimation process in the text. This equation has widely been used in other studies (e.g. Amory et al., 2021; Sauter et al., 2013; Sharma et al., 2023; Vionnet et al., 2014) and therefor we did not consider it necessary to explicitly mention it in the manuscript:
*The last term in Equation 1 accounts for the mass loss of suspended snow due to sublimation based on the formulation of Thorpe and Mason (1966), where sublimation-loss rate of suspended snow is approximated by $\psi_s \phi_s$, with $\psi_s$ as the sublimation-loss rate coefficient.*

> R1-8: **Results:** Results is in my opinion the weakest part of the manuscript. In the current form it seems a disconnected sample of observations and the modelling part with very few connections in both analysis and text. Given the nice dataset that the authors have, and that results are based only in one single case study, I think that authors should extend further the analysis and exploit in more detail the datasets. Some suggestions are:
>
> 1. As the results are based on a single case study I miss a description of the synoptic situation and the characteristics of the front that offer more context to the reader.

The large-scale synoptic situation was described in lines 115-120. We propose to extend the description in the manuscript and will add the following extra information:

- Before the passage of the through, the Alps (and our location of interest) were under the influence of large-scale Southerly flow and moisture transport from the Mediterranean Sea.

- The through axis passed the glacier on Feb 7, 2021 after 18 UTC.

- The associated cold frontal system at the surface likely passed the glacier on Feb 8, 2021, at 09 UTC. The frontal passage is visible in the sudden jump in wind direction in the observations and simulations in Fig.3. Furthermore, the air temperature rises at around 09 UTC (cf. "masked cold front"[1]), in accordance with a decrease in cloud cover (cf. webcam images of 09:00 UTC[2] and 10:00 UTC[3]). An overview of the timeseries of observed and modelled wind direction an air temperature is given in Fig. R2.

We will add a more detailed description of the synoptic situation and the characteristics of the front to Sect. 2.2.

> R1-9: 2. Fig 6: Since Snow redistribution accounts for the snow height on the stations, it would be very interesting to add the observed differences of snow height to panel (a) and compare and show in parallel in
* * *
[1] https://www.meteoswiss.admin.ch/weather/weather-and-climate-from-a-to-z/cold-front.html

[2] https://www.foto-webcam.eu/webcam/hintereisferner1/2021/02/08/0900

[3] https://www.foto-webcam.eu/webcam/hintereisferner1/2021/02/08/1000

panel (a) the results of the observations in snow high and discuss the possible differences. Add also in panel (a) the units.

We will add the units (m) to Fig. 6a. However, Fig. 6a shows the simulated accumulated snow redistribution over the course of 24 hours of simulation time. In Fig. 4, we showed the snow depth observations at AWS28. We do not want to give these different variables in the same plot, as the snow depth observations, besides snow redistribution, include compaction, which would lead to confusion. Fig. 4 only shows the snow depth at AWS28 and we do not elaborate on the snow depths at IHE and StHE, as the snow depth sensors are installed at locations not representative of the surroundings, i.e. at wind-exposed ridges that are almost never covered with snow (also see our answer to R2-12). We stated that already in the results of the original manuscript. In Sect. 3.4, we stated: "However, we compare the snow depth changes from the observations (between 8 Feb, 10:22 UTC and 9 Feb, 01:42 UTC) to the simulated snow redistribution (between 8 Feb, 10:15 UTC and 9 Feb, 00:00 UTC, Fig. 8b). We compare these different variables and see that the order of magnitude of the snow depth changes is twice as large as the simulated snow redistribution due to snow drift, because the observed snow depth changes also include compaction." We plan to add a statement on the observed snow depth changes in relation to the different processes in the introduction of Sect. 3:

*First, the observed snow depth changes from the TLS acquisitions are introduced and discussed in detail. We explicitly note here that the observations from the TLS data only show the snow depth changes, without distinguishing between different processes. Thus, in the next subsections, we explain the driving processes at play with the aid of additional observations at point scale.*

R1-10: 3. The weakest point is that the comparison between simulations and observations is very qualitative (in text and Figure 8), and although the authors give some figures the only conclusion that might be extended from there is that the order of the magnitude of the redistribution in the model agree with observations. I think that a more quantitative approach can be done. First by extending the simulation until 9 February at 2 UTC to include also the 3th TSL acquisition in the simulation. I understand that, as stated in methods, the observations are rescaled to the model resolution (Is this the resolution showed in Figure 8?). If so, the authors can directly compare (1) if at the model resolution the distribution is similar (difference between observations and modeling) and (2) some metrics of the performance of the model such as BIAS, MAE or correlation of the data using the same domain (the one limited by observations). Later can be discussed the role of the compaction on the differences.

We made a more extensive comparison between the model and observations and refer here to comment R1-3. The metrics of the performance of the model are challenging to provide, as ground truth data of of different components causing snow surface elevation changes is not available.

R1-11: Some other minor comments are:
L243-245: The authors use "For scan 1" and "for scan 3", that are snapshots and discuss the precipitation during, before or after. Consider rephrasing with these words instead using "for".

We will reformulate these sentences:
*The precipitation during scan 1 was actually registered after the TLS acquisition, as seen in the 10-min data. Furthermore, we speculate that the precipitation registered during scan 3 is snow drift that is captured by the precipitation gauge.*

R1-12: L247: Figure 6 is shown before Figure 5.

We think that Fig. 6 fits in the Sect 3.2.3. Therefore, we decided we will remove the reference to Fig. 6 in L247 and replace it with:
*In agreement with the observations (Fig. 4), the model simulates precipitation until 06 UTC (see Sect. 3.2.3).*

R1-13: **Discussion and conclusions:** Discussion and conclusions are comprehensive with the results, and correctly details the benefits and the limitations of both the observations and the simulations. However, they are tailed by the qualitative approach in the results. As stated previously, I think that data can be squished to obtain extra insights on the model performance and on the snow redistribution processes during the case study.

We responded to this comment in R1-3 and will adjust the discussions and conclusions accordingly.

R1-14: A small extra point: L341-343. Why is not this stated in methodology?

We will add the following to Sect. 2.3 (Methods):

*They found that the scans have an uncertainty of $\pm 0.10\,m$ in vertical direction after the registration in RiSCAN PRO. In this study, the scans were registered with manually selected tie objects, such as snow-free rocks and the walls of StHE, which led to an better registration than the calculated $\pm 0.10\,m$ in vertical direction with automatically selected tie planes in Voordendag et al. (2023). Additionally to the one-meter grid size DEMs, the high-resolution point clouds are gridded to DEMs with a $\Delta x = 48\,m$ allowing a direct evaluation of the numerical simulations.*

In the discussion we will rewrite the section on the registration to:

*We were able to capture snow fall and redistribution directly thereafter, but we also note that the TLS observations are at the limits of the capabilities of the system. The uncertainty of the TLS observations was estimated at $\pm 0.10\,m$ in vertical direction with manual post-processing in Voordendag et al. (2023). However, the registration is even better in vertical direction if we look at the registration of the scans at the manually selected tie objects in this study. Thus, the snow depth change between scan 2 and 3 was measured reliably and in agreement with observations from the snow depth sensor at the glacier.*

R1-15: **Extra comment:** In author contributions the authors state a contribution of "CS", which is not listed as a coauthor and I assume that is Christina Schmid after reading the manuscript. However, she is not listed as author of the manuscript. I don't know if they forget or not and I cannot evaluate if her contribution is enough for being considered as author, but if not, should be considered take her name off from author contributions and put her instead in acknowledgments.

Christina Schmid will be removed from the author contributions and will be added to the acknowledgements:
*Christina Schmid is acknowledged for the initial model development of the snow drift module.*

**Response to Matthieu Lafaysse**

**Main comments**

R2-1: Voordendag et al. present in this paper simulations of blowing snow over a glacier and its surrounding area (a few km²) with Large Eddy Simulations from the WRF model associated with a new blowing snow module (not fully coupled with the atmospheric model). The simulation period covers only 24 hours including frontal precipitation and a later blowing snow event. Evaluations are done with a few local sensors and with rather unusual Terrestrial Laser Scan measurements available on this area. The paper is well structured and results are correctly described and discussed. Some results are interesting, although it is maybe difficult to identify a really striking conclusion after reading. This might be due to the fact that the general goal of the authors is not really explicit and the reasons for developing a hybrid method for blowing snow simulation (combining very expensive LES and a rather simple blowing snow scheme) are not so easy to understand. As a result, the significance of this paper is difficult to judge as obviously for operational perspectives, the conclusions are very limited by the simulation period and domain size while on processes perspective, the modelling tool might be considered as not ideal compared to previously developed systems. Therefore, I would encourage the authors to more develop their motivations in their revised manuscript.

We thank Matthieu Lafaysse for the careful evaluation of our work. We agree that the motivation of this work should be stated more clearly:

- This work is embedded in the "SCHISM" (Measuring and modeling snow-cover dynamics at high resolution for improving dis- tributed mass balance research on mountain glaciers) project, studying the impact of snow cover dynamics on glacier mass balance - henceforth our area of interest is the Hintereisferner glacier in the Austrian Alps, where its glacier mass balance is monitored for 70 years now.

- To study snow cover dynamics, and especially the spatial pattern, a TLS observational system was installed at IHE, where operational daily scans can be performed. However, studying snow cover dynamics requires more than one scan per day, therefore we decided to perform multiple scans on our selected case study day.

- On the modelling side, the snowdrift module was implemented into the WRF model by Schmid (2021) to study simulated snow cover dynamics with a realistic atmospheric input avoiding simplifications. Until now, the NOAH-MP land model of WRF did not account for effects of wind-driven snow redistribution. Since

our area of interest is located in highly complex terrain, a high horizontal resolution is necessary to achieve realistic simulated wind speeds.

- Therefore, the high-resolution observations are ideal to test the newly implemented snowdrift module in the WRF model in a real setting.

- With the observations-model combination, we try to study the impact of wind-driven snow redistribution on the glacier mass balance - although we are aware that one single case study is not representative for the entire season or years and that further research is necessary.

- With the continuous rise of computational power, an atmospheric model with a snowdrift module can serve as an input for distributed mass balance modelling, such as the COSIPY model Sauter et al. (2020).

We will implement a more clear description of our motivation in the revised manuscript.

> R2-2: Then, as the availability of detailed TLS observations if one of the main strength of this work, I think that the analysis of the simulated patterns of snow redistribution should be the central result of this paper and go further than simple map comparisons. A more detailed and mature analysis of simulation results is to my mind necessary before publication. Beyond these general comments, I also have a number of short comments and questions below that I hope authors can address in their revision process.

We agree with the referee's concern, which has also been brought up similarly by referee #1. We performed a more extensive analysis of the data and added this to the results. For further details, we refer to R1-3.

**Specific Comments**

> R2-3: L35 metamorphism is more common than metarmophosis. I would identify melting and refreezing processes first as phase changes that also induce further metamorphism (but the processes involved are not only metamorphism).

We have rewritten this sentence:
*Compaction of the snowpack can be driven by the overburden of its own weight, the pressure exerted by the wind and/or snow metamorphism processes.*
We have removed "melting and refreezing", as for the scope of our paper the kind of snow metamorphism processes is not of importance.

> R2-4: L63-67 This paragraph is a bit confusing. I think that most LES models can not really be considered as NWP systems as they commonly can not be operated operationally for weather forecasting purposes because of their very high numerical cost. Even if WRF may cover both applications, in the general case this distinction should be better done, especially because the spatial scale of NWP operational systems (at the best kilometric) is not compatible with an explicit representation of blowing snow in mountain terrain. So it is perfectly normal that operational NWP do not represent this process and it should not change in the incoming years. Coupled systems implementing blowing snow between LES atmospheric models and detailed snow models have been developed mainly for processes understanding, but they could not be applied at large scale in NWP applications.

We agree with the referee that operational NWP, even at the kilometric range, are not able to represent blowing snow in complex terrain to a satisfying degree. We did not intend to state that drifting or blowing snow can be simulated at current "NWP resolutions" at the kilometric range. We reformulated the paragraph:
*The numerical simulation of atmospheric processes is usually achieved with numerical weather prediction (NWP) models. Their physics parameterization package usually also includes a multi-layer snow scheme within their land-surface models, e.g. the three layer NOAH-MP scheme in the Weather Research and Forecasting (WRF) model (Niu et al., 2011) or the multi-layer snow model in the Integrated Forecasting System (IFS, Arduini et al., 2019). However, the explicit treatment of wind-driven snow redistribution is often not implemented, because operational NWP models usually operate at the kilometric range, where it cannot be expected that mountainous terrain and small-scale snow redistribution processes are represented correctly. However, at very high resolutions below $\Delta x$=100 m, recent studies couple full stand-alone snowpack models with NWP models. For example, Vionnet et al. (2014)....*

R2-5: L72-73 « the disadvantage of this and other approaches is the extra coupling of the models with a (previously) stand-alone snowpack model. » I don't understand why the fact that these schemes were standalone at the origin would be a disadvantage here. Note also that the numerical cost of detailed snowpack models is actually very low compared to atmospheric models.

This comment is similar to the comment R1-6 by referee #1. We reformulated the statement:
*To our current knowledge, no treatment of snow-driven redistribution is directly implemented in the land-surface scheme of WRF, NOAH-MP (Niu et al., 2011). The advantage of a snowdrift module directly implemented in WRF would be that no extra compilation with additional code is necessary, and that computational costs can be constrained.*

R2-6: L73-81 The proposed methodology is probably useful considering this unusual evaluation dataset, either for process understanding either for model evaluation. However, my feeling is that the governing general motivation of this study is not fully explicit in this introduction, especially considering the concerns I mentioned about the previous paragraph (L63-72). I think the authors could easily improve that point to clarify their motivations.

We thank the referee for this remark - we will add a more detailed description of our motivation in the "main comments" section and will also improve the motivation in the manuscript.

R2-7: L139 Can you provide the surface of the different simulation domains?

The outer two domains utilize ESA-CCI landuse, while we employ CORINE landuse in the two innermost LES domains. The land surface categories in the innermost model domain is "ice" (white surfaces, Figure 1a in the manuscript), while the glacier is surrounded by the land use category "bare rock" in the entire innermost domain.

R2-8: L150 What is the density of fresh snow in the model? Wouldn't it be more realistic to initialize initial density from previous offline simulations of any snow scheme (way less expensive than coupled simulations)? I am afraid the variability of snow density is sufficiently high to potentially be responsible for a bias in initial snow height that could be far from « slight ».

We agree with the referee that a more careful snow initialization would improve the initial snow density in the model. However, this is the first time we test the snow drift scheme in our particular WRF setup, and therefore we try to assess the scheme with WRF's "default" initialization settings. We also discuss the limitations of the snowpack initialization in lines 365-74 in the discussion in the manuscript. A comparison simulation with preliminary (offline) snowpack calculation can be subject to a future model evaluation study and is out of scope for this current manuscript.

R2-9: L173 I don't understand why the ice density appears in Equation 5. First in the current form of the Equation, it could just be simplified. Then, physically, why would the ice density have an impact on the terminal fallout velocity?

This was a typo. We adjusted it and the equation is now the same as in Vionnet et al. (2014):

$$B = \frac{5.516 \cdot \rho_{ice}}{4 \cdot \rho_{air}} \cdot g. \tag{2}$$

R2-10: L188 But does this surface snow density also evolve with snowdrift? And how?

In the model, erosion and deposition do not change the surface snow density. We are aware that in reality this might not be the case. We added an extra sentence explaining this issue to L188.
*Since the particle erosion process depends on the cohesive bonds of the snow particles, the surface shear stress threshold is a function of the snow density $\rho_s$, which is not affected by snow drift in the model (Walter et al., 2004).*

R2-11: L211 « the slopes adjacent to HEF showed a more heterogeneous snow distribution between scan 1 and 2 (Fig. 3a), which might indicate snow redistribution during the snow fall event ». I am not sure where to look at but it is honestly hard to distinguish any spatial variability in Fig. 3a. Then what are the arguments

to identify this variability as snow redistribution during the snowfall event? i.e. how to disentangle from local scale precipitation variability?

The heterogeneous snow redistribution at the slopes adjacent to HEF is for example observed around StHE and depends on the local topography. The variability in the snow depth might indicate snow redistribution, but also, as indicated by the referee, preferential snow deposition (Mott and Lehning, 2010). We reformulated this sentence: *The snow was evenly distributed over the glacier surface, but the slopes adjacent to HEF showed a more heterogeneous snow distribution between scan 1 and 2 (i.e. around StHE, Fig. 3a), which might indicate preferential snow deposition and/or snow redistribution during the snow fall event (e.g. Mott and Lehning, 2010).*

R2-12: L215 It is a bit surprising to have sensors installed in unrepresentative locations and that the high resolution of TLS would not be able to identify the mentioned local anomalies. Although the authors explain they do not want to consider data assumed to be inconsistent at the IHE and StHE sites, the results are shown in the Figure and their discrepancy with TLS measurements raises questions.

In case of the two stations StHE and IHE, both were not installed for the sole purpose of the present snow redistribution study. However, since the data was available for our time period of interest, we briefly analysed the data. We have to clarify the choice of locations of the stations:

- StHE is located close to University of Innsbruck's mountain shelter to ensure accessibility during harsh weather conditions. The station is operational since 2010 and its focus is laid on long-term observations of atmospheric conditions close to the glacier (Strasser et al., 2018); before 2010, the station mostly measured during summer and its data have been used for more than 40 years (e.g., Obleitner, 1994).

- The station IHE is located at a mountain ridge to gain more insight on the interaction between the free atmosphere and the mountain boundary layer close to the glacier. The turbulence flux tower allows detailed boundary-layer studies, such as model evaluation (Goger et al., 2022) and testing of turbulence theory (Stiperski et al., 2021; Stiperski and Calaf, 2023).

Snow depth on wind exposed ridges is a function of snow fall, wind redistribution and underlying terrain micro relief. On StHE the micro relief at the weather station consists of irregular blocks and ridges up to 1 m high interspersed with shallow depressions about twice their width. Early winter snow falls cover the depressions and smooth the terrain, but most tips of boulders and ridges remain snow free, which causes very noisy snow depth data. Additional snow falls increase snow depth only for short periods (if at all) until wind erodes and redistributes further (Fig. 3c). On the ridge where IHE is placed, the wind forms a snow cornice of several meters in thickness directly next to permanent snow free areas (where unfortunately also the snow depth sensor is installed). Also, the snow depth at IHE is not visible in the TLS analysis as the weather station is located behind and above the TLS station.

Because in this study we concentrate on the smooth terrain on the glacier, we qualify the two locations StHE and IHE as unrepresentative for snow cover dynamics at the glacier. However, the observed wind speed and direction at these stations give a good overview of the meteorological situation at the glacier's surroundings. We clarified the wording in L215 of the original manuscript:
*In this study, we do not elaborate on the snow depths at IHE and StHE, as the terrain-dependent snow cover dynamics are unrepresentative at these two stations compared to the rather smooth and homogeneous glacier surface around AWS28.*

R2-13: L236 « Melt can be excluded during the cold case study period » I guess this is true but the fact that temperatures are cold is an insufficient argument to exclude melt at all slope aspects. The result of a simulated surface energy balance would be a better argument to exclude melt.

Melt requires surface temperatures at 0°C and additional energy input to supply the phase change from solid to liquid. Surface temperature measured at AWS28 are always well below 0°C (Fig. R3 and thus, melt can be excluded. We assume this also holds for the rest of the observed area. We clarified this in L236:
*Melt can be excluded due to surface temperatures persistently below the freezing point.* Because we lack the observations from the surrounding slopes, we checked the simulated skin temperature in the model, and they are consistently below freezing point during the entire simulation period.

[Figure]

Figure R3: Surface temperatures at AWS28 during the case study period.

R2-14: L245-249 The authors say that « snow redistribution patterns can only be simulated correctly if the modelled precipitation and wind patterns agree with the observations. » However, there is not any attempt in the paper to evaluate the uncertainty of simulated precipitation. It is true that measurements are affected by undercatch. However, correction functions exist to account for undercatch (e.g. Kochendorfer et al., 2017). Precipitation from atmospheric models are also known to be prone to very large errors. Therefore, as data are available, I would highly recommend to include in the paper the comparison between observed and simulated precipitation. I could imagine that this work was actually done by the authors and excluded from the paper because of large discrepancies between simulations and observations. However, being aware of these discrepancies is to my mind very important to have in mind the uncertainty of precipitation amount.

We agree with the referee that a more detailed evaluation of the simulated precipitation with observations is necessary. We used the glacier winter mass balance data as proxy for winter precipitation for the winters 2020/21-2022/23 and found that precipitation is underestimated by 30/50%. Correcting for undercatch using Kochendorfer et al. (2017), precipitation is overestimated by 10-16% for the same winters. We also added the precipitation observations from StHE to Figure 6 in the manuscript (Fig. R5). During the simulation period, modeled and observed precipitation agree well both in timing and amounts. One of the differences is that the precipitation stops earlier in the model than in the observations. Taken into account that observed and modelled precipitation are two different physical quantities by the way they are obtained, we conclude that the model is able to simulate the precipitation pattern on the case study day accordingly. We added the description of the precipitation time series in the manuscript.

We also note that the precipitation at IHE is not analysed in the manuscript, because the focus of the station is on a different topic as outlined in R2-12. The precipitation gauge is placed at a wind-exposed ridge that is hardly ever covered with snow and thus prone to large amounts of undercatch. For the scope of this review, we plotted the precipitation data of IHE to Fig. 4. Only 18 mm total precipitation was registered at the ridge station IHE, whereas this was 56 mm at StHE (Fig. R4). The precipitation gauge at IHE is placed directly at the ridge and was placed there as a manner of experiment, as this was a different type of precipitation gauge (Geonor) than the other Ott sensors in the catchment. The surface below this precipitation gauge is almost never covered with snow and

[Figure]

Figure R4: Precipitation (StHE: orange; IHE:blue) and relative snow depth observations (AWS28: pink) during the case study period. Note that the precipitation observation is not corrected for undercatch and the snow depth is arbitrarily chosen to start at 0 m at the minimum observed during the case study period, despite the glacier was already covered in snow and thus, the actual snow depth was more than 0 m. The grey bars indicate the time of the TLS acquisitions.

therefore extremely prone to undercatch. The precipitation data of IHE is not mentioned in the manuscript, as this would only lead to confusion about the differences between StHE and IHE.

We will add a description of the comparison between the observed and modelled precipitation at StHE in the revised manuscript.

> R2-15: L264-265 « Overall, the model simulates the wind field on the glacier and the surrounding with confidence and provides good input conditions for the snow redistribution and snow water equivalent (SWE) estimates. » Confidence is always relative. I think that this sentence could me more quantitative (e.g. providing an error metric). Furthermore, Figure 5 allows only to evaluate the ability of the model to simulate the wind temporal variability, but the ability of the model to simulate its spatial variability can not be really estimated while the wind spatial patterns are especially important in snow redistribution. This should encourage to moderate this conclusion.

We agree with the referee that with our current observations available, we cannot judge whether the model is able to simulate the spatial pattern of wind speeds and directions correctly. However, we have three stations at three different locations (glacier tongue, South-facing slope, and a mountain ridge), and we noted from previous studies (e.g. Goger et al., 2022), that they are located at representative locations to discuss the structure of the larger-scale flow (IHE located at the mountain ridge), or the major wind speed/direction over the glacier (AWS28). We calculated Bias and Rmse values for wind speed for the three stations with available measurements (Table 1), and note a generally satisfying model performance: The smallest model error is present at the glacier tongue (AWS28) while the general wind speed is overestimated by around $2\,\mathrm{m\,s^{-1}}$ by the model for both StHE (slope) and IHE (mountain ridge). We added the table and the description of the model error to the manuscript.

| Station | Bias | Rmse |
|---:|:---:|:---:|
| StHE | 2.35 | 3.53 |
| IHE | 2.08 | 4.23 |
| AWS28 | 0.96 | 1.81 |

Table 1: Bias and Rmse calculated from station data for the wind speed over the period of the simulated 24 hours.

[Figure]

Figure R5: Time series at 8 Feb 2021 from the closest grid point in the model to StHE (orange), IHE (blue), and AWS28 (pink) in panel a) of accumulated snow redistribution (relative to the start of the simulation at 00 UTC), and panel b) modelled (StHE and IHE) and observed (StHE) solid precipitation (bars), and the relative change of snow water equivalent (Δ SWE, lines).

> **R2-16:** L267-279 Is compaction represented in the 3-layer snow model? In that case, would it be possible to compare the simulated compaction for this specific event (e.g. disactivating snow drift) with the simulated compaction of the 2021-2022 and 2022-2023 seasons in offline simulations? It may allow to better characterize this specific situation than considering all observed compactions over these 2 years.

Yes, compaction is accounted for in the 3-layer snow model, as outlined in the discussion of the original manuscript (lines 370-375). We described in the discussion that the calculation of compaction is challenging, as the snow pack is initialized as a snow pack with only fresh snow, whereas it is a layer of old snow, with fresh snow on top in reality. However, compaction is taken into accounts by the model, but strongly overestimated as the initialized snow pack has a small bulk snow density. The computationally expensive model setup does not allow for a correctly initialized snow pack and thus, only simulating compaction will lead to similar results: overestimation of compaction of an entirely fresh snow pack.

> **R2-17:** L284-285 The sign should be given (i.e. erosion or accumulation), I guess erosion from the plot but it should be explicit in the text. The unit is also missing on the top panel of Fig. 6. Then, why expressing this value in terms of height rather than in terms of mass? As blowing snow also modifies the density, it is bit counter-intuitive to mix mass transfer and density changes in a single diagnostic of snow height change, especially if the authors want to interpret this result in terms of mass change as in line 289.

This was indeed negative snow redistribution (erosion). We have changed it accordingly in the text:
*The simulated snow redistribution is found to be -0.022 m at IHE, -0.014 at StHE, and -0.003 m at AWS28 at the end of the simulation period (Fig. 6a).*

> **R2-18:** L299 « SWE remained constant until 15 UTC due to compaction » : snow depth reduces because of compaction, but compaction does not make the SWE to not change.

We reformulated this sentence:
*To summarize, SWE increased until 06 UTC due to solid precipitation, while SWE remained constant afterwards and until 15 UTC as only compaction took place. After 15 UTC snow redistribution led to a continuous decrease in SWE at all stations.*

R2-19: L300-304 Was there any evaluation of simulated sublimation with this blowing snow scheme in previous studies? How reliable is this very low estimation of sublimation?

The snow drift scheme was first tested by Sauter et al. (2013) on the Vestfonna icecap, Svalbard, but with a different model. Therefore, the current work is the first study in real-case simulations with this snow drift scheme, therefore we do not yet have options for comparison with previous studies. We use the same calculation for the sublimation as in previous studies (Thorpe and Mason, 1966), therefore we think that the calculated values are not unrealistic. Please also refer to our comment to R1-7.

R2-20: Figure 7 : As mentioned before, I am a bit surprised about the choice to present erosion/accumulation in meters and not in kg/m² as it does not allow to separate changes in density and changes in mass, and as in this Figure there is no comparison with observations.

Snow redistribution is presented in meters to be able to compare it to the TLS data. We agree that there is no comparison with the observations in this figure, but the aim of this figure was to show the time evolution of the snow redistribution. Also, we wanted to be consistent throughout the manuscript. We added this information to the Figure caption.

R2-21: L310-311 « The simulated flow patterns were, in agreement with the observations, almost perfectly down-glacier (Fig. 5). » I don't understand this statement, could you please rephrase?

We rewrote the sentence to:
*The simulated wind direction was mostly down-glacier, in agreement with the observations from AWS28.*

R2-22: L311-321 So I understand that the model is not mass-conservative across this small domain as erosion is much higher than accumulation and sublimation is negligible. That means that blowing snow fluxes at the domain boundaries are especially important while there is a discontinuity in resolution at this boundaries. First of all, was mass conservation checked at larger scale (on the coupled domains)? Then, how boundary fluxes may be impacted by this discontinuity in resolution and could it have an impact on the resulting unbalance between erosion and accumulation on the studied domain?

We thank the referee for the raised concern. The model is not mass-conservative, since we cannot account for how much snow mass is advected into the domain and how much is advected out. The discontinuity in resolution at the boundaries is not only relevant for blowing snow fluxes, but for any other prognostic variable in the domain as well. Therefore, several solutions exist in WRF to tackle this problem:

- At first, a reasonable nesting ratio between domains is chosen. In WRF, usually values of a factor 3 or 5 are chosen. In our case, we chose factor 5, in accordance with other numerical studies over complex terrain (Umek et al., 2021; Gerber et al., 2018; Goger et al., 2022).

- WRF uses a lateral boundary zone (10 grid points in our case), where topography is smoothed to ensure numerical stability.

- The numerical setup required a careful placement of the domains to avoid numerical instabilities (Goger et al., 2022, their model decription section), and due to the complex topography, we ensured a domain placement to ensure realistic upstream conditions (i.e., no mountain ridge close to the domain borders to avoid unrealistic wind speed-ups at the ridge which would travel into the domain).

- Our innermost domain is centered over the glacier and spans 250x250 grid points. resulting in 12 km in each direction. In case of small inconsistencies, we assume they dissipate until they reach our area of interest.

- Still, we plotted cross-sections of snow particle concentration for both North-and-South and East-and-West cross-sections with `ncview` at 16 UTC (when snowdrift is present) in Fig. R6. If large inconsistencies in particle concentration were present, it would be visible at the first and last ten grid pints of the domain.

Given this, we assume that erosion dominates over accumulation in our simulations, mostly attributed to the high wind speeds (more than $10\,\mathrm{m\,s^{-1}}$) in the model. The high winds speeds were also observed (cf. Fig. 3 in the manuscript), and the TLS acquisition also suggests only small, negligible snow accumulation zones - while wind erosion (and compaction) are the dominant processes.

[Figure]

Figure R6: Snow concentration from the lowest model levels for a South-North (left panel) and West-East (right panel) cross-section from 16 UTC.

R2-23: L323-332 The reasoning about the average magnitude of compaction and snow redistribution to explain observed snow height changes is interesting. However, the interest of such detailed modelling in a context with high resolution observations is obviously to look at the realism of spatial patterns. In this section, the evaluation of simulated spatial patterns remain unfortunately rather subjective with a rather vague map comparison. To my mind, this is disappointing. I would have expected a much detailed spatial analysis exploring spatial correlations between simulations and observations or other metrics allowing a more objective evaluation of the simulated spatial patterns.

The high-resolution TLS scan ($\Delta x$ =1 m) reveals a detailed structure of snow cover redistribution, including the snow redistribution from the windward to the leeward side (Fig. 3c). However, when the high-resolution TLS data is upscaled to $\Delta x$=48 m, many of the detailed structures, especially at the glacier, disappear at the model's resolution. Furthermore, mountain ridges, where most of the snow erosion takes place (according to the model and webcam images), are not covered by the TLS data. Therefore, a direct comparison of spatial patterns is a challenge - and from the area covered by the TLS (mostly the glacier), a spatial autocorrelation analysis would not reveal new insights. However, we performed a comparison between the single grid cells of the upscaled TLS data and the model output and produced a scatter plot (refer answer to R1-3). This plot revealed that the model underestimates the amount of snow redistribution by 9.1% and that 6.4 cm of the surface elevation change can be attributed to compaction, which corresponds to our estimate with the SWE and snow depth data of the preceding years. We will indicate in the revised manuscript that detailed structures of snow redistribution as seen in the high-resolution TLS data is not visible anymore in the upscaled data at $\Delta x$ =48 m and that this challenges a comparison between the observations and the simulations.

R2-24: L347-348 « Therefore, we assume that no model bias emerges due to erratic wind patterns » Authors should specify that this statement only applies for their specific study case as there is no evaluation of systematic biases on longer periods.

We will add the following sentence:
*Still, we have to keep in mind, that this very satisfying model simulation only applies to our case study and could look different for other time periods or locations.*

R2-25: L348 « The simulated snow redistribution is realistic in terms of spatial structure and magnitude. ». This strong conclusion should be supported by more objective results than simple maps comparison as previously noticed.

We will rewrite the sentence to a more cautious formulation in the manuscript. We agree with the referee that a more detailed analysis of our results is necessary. We already attempted this in R1-3 and additionally we calculated the difference between the surface elevation changes of the upscaled TLS acquisitions and the modelled snow drift redistribution (Fig. R7). Please keep in mind that the observations still include the compaction. At a first glace, the model generally underestimates the surface elevation change on the entire domain (this is in agreement with the findings in R1-3). However, it is evident that the largest changes occur at the slopes surrounding the glacier, this can be mostly explained due to information loss in the upscaling of the TLS acquisition, and unresolved topography

[Figure]

Figure R7: Difference between the surface elevation change ($\Delta H$) as derived from the TLS data and the simulated snow redistribution.

in the model. On the glacier surface, the difference between model and observations is reduced. This suggests that the glacier surface is already well-resolved in the model, because the glacier surface is rather flat compared to the surroundings. This is good news for the possible next step of this analysis - namely, assessing the impact of wind-driven snow redistribution on the glacier mass balance. We will add Fig. R7 and the accompanying discussion to the manuscript.

We calculated the spatial autocorrelation function for model topography and the accumulated change in snow height due to snow redistribution (Fig. R8). The spatial pattern of the two fields clearly align, allowing the conclusion that the wind-driven snow redistribution is strongly dependent on the topography in the model. We tried to produce the same figure with the TLS data. Unfortunately, due to limited coverage of the domain, spatial autocorrelation can not be calculated due to missing number values. This is clearly a weakness in using TLS data for spatial model evaluation.

[Figure]

Figure R8: Spatial autocorrelation function for model topography (upper panel) and the snow height change due to snowdrift (lower panel) after 24 hours of simulation time.

> **R2-26:** L349 Again the 'smoother' behaviour of simulated snow fields should have been quantified for instance by a spatial variance analysis of the observed and simulated maps.

Fig R1 revealed that the observations are noisy compared to the simulations, please also refer to comment R1-3. The model is not able to simulate processes such as avalanches and also the model topography can not incorporate for steep cliffs where no snow is accumulated, but this is only observed in the high-resolution TLS data ($\Delta x = 1$ m).

> **R2-27:** L349-350 Although this is a plausible explanation, I think there is not in the paper a rigorous demonstration that this process is able to explain the spatial variance discrepancies between model and observations and that only this one is involved.

We agree that the smoothed model topography might not be the only reason for possible discrepancies between the model and the observations. We again refer to the scatter plot and the answer to R2-26.

> **R2-28:** L359 I think I don't really understand why simplicity in the snow drift module and snow scheme is really an advantage when the choice for the atmospheric model is a very expensive LES model that can only be applied for very short simulation periods and that in any case prevail in the numerical cost. This is probably linked to the lack of clarity of the governing objective of this study, is it a process study, a model evaluation, something else?

We will clarify our motivation in the revised version of the manuscript and will add an explicit statement on the novelty of the work. This was the first time the snowdrift scheme was applied to a real-case WRF simulation, and the TLS acquisitions posed an unique opportunity to evaluate the scheme with high-quality data. However, to represent the atmospheric processes in highly complex terrain accordingly, a very high horizontal resolution is necessary in the numerical simulation, since at least ten grid points are necessary across a valley for the correct representation of atmospheric processes (Wagner et al., 2014). Currently, it is not intended for the scheme to be implemented in an operational forecasting processing chain, but it can be useful for selected case studies.

> **R2-29:** L371-373 « The results of this snow compaction (not shown) are overestimated, because the model assumes the entire snowpack (>2 m) to compact and not only the 0.48 m of fresh snow. » This statement is very unclear and obviously compaction happens in the whole snowpack. Please rephrase.

We will rephrase the sentence:
*The results of this snow compaction (not shown) are overestimated, because the model is initialized with a snowpack entirely consisting of fresh snow (>2 m of fresh snow), enabling high compaction rates, whereas in nature there is only the 0.48 m of fresh snow on top of older snow layers available for compaction.*

> **R2-30:** Finally a section dedicated to data availability is necessary to fit data policy for publication in The Cryosphere: `https://www.the-cryosphere.net/policies/data_policy.html`.

Thank you for the remark, we will adjust the data availability section accordingly.

**References**

Amory, C., Kittel, C., Toumelin, L. L., Agosta, C., Delhasse, A., Favier, V., and Fettweis, X.: Performance of MAR (v3.11) in simulating the drifting-snow climate and surface mass balance of Adélie Land, East Antarctica, Geoscientific Model Development, 14, 3487–3510, doi: 10.5194/gmd-14-3487-2021, 2021.

[revised manuscript text omitted]

---

## Referee Report (RR1)

Authors have correctly addressed most of my comments. Now the manuscript benefit of the publication of the snow drifting module and on a more quantitative analysis of the case study. I want to also recognize the effort of the authors on calculating the compactation, to better evaluate the drifting model. I think that this paper is now acceptable to be published in TC after addressing some few minor comments more that arised in this second lecture of the paper.

L69: accordingly Wagner et al. (2014)

L124: southerly without caps

L125: If the trough is over France, are you sure is moving westward? Or is it eastward?

L170: there is a problem here with the sentence going beyond the margin and the end of the page.

Figure 8. For better comparison I would suggest to represent b and c panels only for those points with data in a. In the current figure I find difficult to compare the points between a and b.

L387: line typo

L412: I think that this sentence does not totally agree with the results showed.

- Only large-scale spatial structure is realistic (L390: model is not able to capture the small-scale snow depth structure at the slopes), and it links with the following sentence.

- Magnitude is not realistic (L372: The order of magnitude of the snow depth changes from the observations is twice as large as the simulated snow redistribution due to snow drift from the simulation).

Maybe the authors can rephrase lowering the claim.

---

## Author Response (AR3)

**Response to Referees**
**Investigating wind-driven Snow Redistribution Processes over an Alpine Glacier with high-resolution Terrestrial Laser Scans and Large-eddy Simulations**

Annelies Voordendag, Brigitta Goger, Rainer Prinz,
Tobias Sauter, Thomas Mölg, Manuel Saigger and Georg Kaser

December 21, 2023

Dear editor and referees,

We would like to thank the editor for handling our manuscript and the referees for their careful evaluation of our work and the valuable suggestions, comments and questions. We believe that the manuscript substantially benefits from the referees' feedback. Below we address our detailed responses to all the comments.

In this response-to-review document we try to clarify and address each of the suggestions, comments and questions made during the review. Therefore we have copied the comments in blue boxes and have addressed them one by one. In the response we use italic fonts to quote text from the revised manuscript.

Yours sincerely, Annelies Voordendag, Brigitta Goger & co-authors

**Response to referee #1**

**Overview**

> R1-1: Authors have correctly addressed most of my comments. Now the manuscript benefit of the publication of the snow drifting module and on a more quantitative analysis of the case study. I want to also recognize the effort of the authors on calculating the compaction, to better evaluate the drifting model. I think that this paper is now acceptable to be published in TC after addressing some few minor comments more that arised in this second lecture of the paper.

We are very thankful for the constructive comments of referee #1.

**Piecemeal**

> R1-2: L69: accordingly Wagner et al. (2014)

The sentence now reads:
*At this resolution, both topography and glacier ice surfaces in the Alps can be expected to be well-resolved, given that at least 10 grid points across a valley are necessary to resolve the relevant boundary-layer processes (Wagner et al., 2014).*

> R1-3: southerly without caps.
> R1-4: L125: If the trough is over France, are you sure is moving westward? Or is it eastward?

Corrected these typos. The sentence now reads:
*The southerly flow was mostly associated with a trough over France moving eastward towards the Alps, while the trough axis passed our location of interest on 7 Feb 2021 after 18:00 UTC.*

> R1-5: L170: there is a problem here with the sentence going beyond the margin and the end of the page.

This is a problem with *latexdiff*. In the version without track changes, this problem is not apparent.

> R1-6: Figure 8. For better comparison I would suggest to represent b and c panels only for those points with data in a. In the current figure I find difficult to compare the points between a and b.

We added grid lines to all subplots in Figure 8 and the glacier outlines as used by the model in Figure 8a. Removing simulated data in Figures 8b and c would limit us in our statements on the good model simulations at the ridges, as the ridges are sparsely captured by the TLS system, but realistically simulated in the model.

> R1-8: L387: line typo

This is a problem with *latexdiff*. In the version without track changes, this problem is not apparent.

> R1-9: L412: I think that this sentence does not totally agree with the results showed.
>
> - Only large-scale spatial structure is realistic (L390: model is not able to capture the small-scale snow depth structure at the slopes), and it links with the following sentence.
>
> - Magnitude is not realistic (L372: The order of magnitude of the snow depth changes from the observations is twice as large as the simulated snow redistribution due to snow drift from the simulation).
>
> Maybe the authors can rephrase lowering the claim.

L412 (from the manuscript with track changes) has been clarified:
*The simulated snow redistribution is realistic in terms of spatial structure and magnitude. However, the processes at smaller scales are smoothed out, which is due to the horizontal resolution of 48 m and the smoothed model topography restricted by numerical stability. The model topography limits the slope angles to a maximum of 35°, and thus the model topography clearly deviates from real topography.*
The magnitude of the snow redistribution is realistic and we also stated several times that the surface elevation changes are not directly comparable to the snow redistribution simulations, as the surface elevation changes also include compaction. For example in L373/374 directly after the statement in L372 and earlier in L222/223. This has been elaborated in the description of Figure 9.

**Response to Matthieu Lafaysse**

**General comments**

> R2-1: I thank the authors of this paper for their detailed answers to my comments and their attempts to introduce new results in the paper to strengthen its conclusions, regarding precipitation and wind forcing, and mainly concerning the spatial variability of snow depth changes which is the main interest of using these TLS data. However, I feel when reading that this new analysis is maybe not yet completely mature (see my detailed comments).

Thank you very much for the positive feedback and for reviewing our manuscript again.

> R2-2: The goal and impacts of the study have also been clarified. I think that additional information about the implications of coupled and uncoupled processes in this system and about the numerical cost would help the community to identify the pros and cons of this approach compared to more expensive or cheaper simulation systems for their application. In particular, the authors suggest in their answer that the feasibility of applying such model over a full season is not so far, but there is nothing in the paper that support and explain that if this is true. On the contrary, a large part of the discussion is dedicated to the implication of not being able to run a continuous simulation to get realistic initial snow density. Therefore, the readers need to understand why the high numerical cost dedicated to the atmosphere is justified compared to the numerical cost dedicated to snow processes for this kind of analyses.

We are - indeed - not sure whether the current set-up is applicable to entire seasons. If seasonal simulations are the goal, they could be also run with a coarser grid spacing (e.g., $\Delta x = 240\,\mathrm{m}$). However, it has to be considered then that the topography and boundary-layer processes are less well resolved than at $\Delta x = 48\,\mathrm{m}$. We treat this point in the discussion.

R2-3: The limitations of the single-case study are well described in the discussion. The code and data availability section has now been provided. I don't know the reasons to not share the TLS data (technical or political) but I believe this kind of data set could also be highly valuable for the community if available, although it is probably not mandatory for this journal.

The TLS data has so far not been published as we wanted to wait until the PhD of A. Voordendag was finalized. Also, there are several products available from the TLS data (registered/unregistered point clouds, digital elevation models) and with 3 years of (nearly) daily data, we have not yet decided on an appropriate way to publish the large data set. However, the data is available upon request from ACINN/Rainer Prinz.

R2-4: Although I think that a second round of improvements is necessary before publication, I recommend the authors to finalize this promising work as detailed evaluations of snow transport models are challenging and insufficient in the current snow modelling literature.

Thank you again for these constructive comments.

**Detailed comments**

R2-5: The line numbers in my report correspond to the manuscript with tracked changes.

Thank you for the clarification.

R2-6: L52 validate → evaluate

Changed.

R2-7: L53 The reason to cite SNOWPACK is unclear. It would be better to mention the large variety of available snow models in the literature (Krinner et al., 2018; Menard et al., 2021). Then, snow processes are also very commonly simulated in coupled mode. Maybe the goal of limiting this sentence to standalone simulations was to be more specific about higher resolution simulations? If yes, it should be said.

We explicitly mentioned SNOWPACK because it was recently coupled to the atmospheric model we are using in our study (WRF). We added the inter-comparison studies suggested by the referee and re-wrote:
*Modelling snow processes is usually achieved by a large variety of standalone snow models, which receive input data from atmospheric models or observations (Krinner et al., 2018; Menard et al., 2021). Recent studies also couple full (previously) stand-alone snowpack models with atmospheric models.*

R2-8: L59-60 «CRYOWRF can successfully simulate snow accumulation »and redistribution both over the Swiss Alps and Antarctica (Gerber et al., 2023).» I have checked this reference. It only presents simulations over Antarctica, and it does not demonstrate that «snow accumulation and redistribution are successfully simulated». (It depends what is supposed to mean «successfully», I guess it was consistently with observations). This statement should be reformulated closer to the actual conclusions of this paper (the evaluated variables are local-scale blowing snow occurrence and local-scale surface mass balance, but not the redistributed snow mass). The evaluation of snow redistribution is extremely challenging for all simulation systems and I think conclusions should always be formulated with caution and accuracy.

This was a mistake from our side - we wanted to cite Sharma et al. (2023), where the simulation over the Alps is explicitly highlighted (their Figs. 15-17), and also mention the applicability over Antarctica with Gerber et al. (2023). We re-wrote the sentence:
*First results suggest that CRYOWRF is capable of simulating snow accumulation and redistribution over the Swiss Alps and Antarctica (Sharma et al., 2023; Gerber et al., 2023).*

R2-9: L61 I don't think that «golden standard» is an appropriate expression. The choice of numerical models depend on applications and CryoWRF and Méso-NH-Crocus are definitely not a golden standard for instance for real-time operational snow simulations designed to monitor snow cover over large domains, or for coupled climate models designed to be able to run over the whole century. However, these models are indeed the ones resolving the most in detail all the coupled physical processes of blowing snow, and this is the best choice for

> process studies on dedicated case studies, at the expense of a very high numerical cost.

We re-wrote the sentence:
*While fully coupled snow-atmosphere model chains likely resolve coupled processes and atmosphere-cryosphere interactions well for case studies (at a high numerical cost),*

> **R2-10:** L66 What does mean «relevant» here? Relevant for which application?

Changed to "relevant for process understanding".

> **R2-11:** L72 «In contrast to coupled modelling systems» would suggest that the snow transport module used in this work does not have feedback to the atmospheric model whereas if I understood well (from L171 and L175), it does. Please rephrase to clarify.

We agree and removed the formulation.

> **R2-12:** L72 I don't think that the absence of change in the compilation procedure is a strong argument to justify the interest of the approach.

We agree and wrote a different sentence:
*To our current knowledge, this is the first time where an openly available, easy-to-use formulation for wind-driven snow redistribution is implemented in the WRF model code.*

> **R2-13:** L71-78 Although the proposed modifications improve the understanding of the objectives of this paper, I think that the target applications of this numerical system are still not explicit enough in this paragraph. «Study the impact of wind driven snow redistribution on a large Alpine glacier for a case study» is the goal of this study, ok. Is it the main application of this modelling system? Or does it have broader objectives?

This study is the first one to test and evaluate the snowdrift module after the initial implementation for a real-case study. Given the TLS observations and the complex environment, the glacier is an ideal test-bed to assess the model performance (for both the atmosphere and cryosphere component). We changed the sentence to:
*We present a first evaluation of the newly implemented snow drift scheme with high-resolution TLS observations and examine whether the model delivers realistic results in snow depth change and spatial patterns in this highly complex environment.*
The broader objective of the snow drift module in WRF is to add an easy-to-use formulation of wind-driven snow redistribution in the WRF model.

> **R2-14:** L77-78 As the study is limited to a specific event, is it really possible to estimate the impact on the glacier mass balance? At least, it is unclear at this stage why it would be possible. And finally line 443, the authors acknowledge this is not possible.

Exactly - we check whether the set-up bears the potential to asses the impact on the mass balance and then conclude that currently it's not possible. Still, the study shows us a first estimate *whether* the model (and observations!) are able to detect wind-driven snow redistribution in a reasonable way. However, we added in the conclusions that the computational power restraint is obviously limiting the study.

> **R2-15:** L171 Here, the «coupled» word means that there is feedback to the atmosphere, right?

Exactly - we added the feedback in brackets.

> **R2-16:** L192 This is a constant for pure ice density, right? This should be specified to avoid confusion with glacier ice density.

Indeed - changed.

> **R2-17:** L208 Is $\rho_s$ the density of the upper snow layer? Please specify. Also I realize it is not clear how the 3-layer snow model simulates compaction and as I have already asked during the first review, what is the density of new falling snow? These components of the model have to be detailed when the snow model is

[Figure]

Fɪɢ. 1. Plot of bulk snow particle density vs diameter showing disdrometer observations of Brandes et al. (2007) and the modeled relationship using assumptions in the new bulk microphysics scheme. The typical value used in most models is a constant (0.1 g cm$^{-3}$).

Figure R1: Snow density as simulated by the Thompson scheme (green line). Taken from Thompson et al. (2008, their Figure 1).

introduced in Sect. 2.4, so that it is possible to understand their interaction with the snowdrift module.

Yes, $\rho_s$ is the density of the upper snow layer. The density of the falling snow is described in Thompson et al. (2008), where snow assumes a nonspherical shape with a bulk density varying with diameter (Fig. R1).

The alternation of the snow surface structure and snow density by erosion and deposition can cause feedbacks onto the drifting snow and flow field structure is currently not included in our scheme - however, it is planned to be added later (M. Saigger, personal communication).

We mentioned in the discussion of the revised manuscript (line 433) that snow compaction in NOAH-MP is included following the empirical formulations by Anderson (1976), while snow compaction by the snowpack's own weight is calculated after Sun et al. (1999). We agree that we have to introduce the compaction description earlier, therefore we added it together with the references in Section 2.4 (Numerical Model).

R2-18: L256 Are simulated surface temperatures below freezing point over the whole glacier? If yes, this spatial extent of the statement should be mentioned.

Changed sentence to:
*[...], the simulations also suggest that surface temperatures remain below freezing point over the entire glacier.*

R2-19: In Figure 3, the color contrast is low between the orange point and the map color scale in Fig b and c, while it is important to distinguish the point to understand the comments Lines 232-233.

We agree and added black circles around the station locations to make them easier to distinguish from the surroundings. We also adjusted this in Figures 8 and 9.

R2-20: L322-323 «Taken into account that observed and modelled precipitation are two different physical quantities by the way they are obtained» I don't understand what the authors mean. Simulated and observed weather or snow variables are always obtained differently but we still have to evaluate the reliability of simulations. What is specific to precipitation here? Why would observed and simulated precipitation not

[Figure]

Figure R2: a) Observed snow depth changes over HEF at $\Delta x=48\,$m between 8 Feb, 10:22 UTC and 9 Feb, 01:42 UTC plotted against the simulated snow redistribution for all the covered grid cells (blue) and the linear fit between these variables (black) **with harmonized x- and y-axis**. b) The difference between the observed snow depth changes and the simulated snow redistribution over the region of interest.

> represent the same physical quantity? I would understand from Fig 6 that new snow accumulation is probably underestimated by the model.

This sentence was unnecessarily complicated - we re-wrote it:
*We conclude that the model is able to simulate the temporal pattern on the case study day successfully, albeit with a slight underestimation.*

> R2-21: L324 temporal pattern?

See above.

> R2-22: Figure 9a: To better identify the agreement or not between observations and simulations, can you harmonize the x and y axis boundaries so that it looks like a traditional scatter plot ? I also understand from this plot that the observed and simulated variabilities highly differ. This is not so surprising but this should be commented. Is there a significant correlation in this plot ? (you could provide the value of R2). Note that this kind of pixel-to-pixel evaluation is extremely challenging for any snow transport model, and I would not be surprised that the agreement could be moderate or low. It still worth showing this kind of result to be aware of the limitations of snow transport models.

Figure R2 shows Figure 9 from the manuscript with harmonized x- and y-axis. As pointed out by the reviewer, the variability of the observations and simulations differ strongly. The simulations only range between -0.073 and -0.0032 m, whereas observations are found between -2.82 and 3.9 m. These variations in the simulations can be attributed to processes such as avalanches and small-scale snow redistribution from the windward to the leeward side. Due to these variations, the coefficient of determination ($R^2$) is found to be 0.005 and as expected by the reviewer, this is low. We decided to remain with the figure as presented in the original manuscript, but note that we added grid lines and the glacier outlines as used by the model to Figure 9a.

We have rewritten L386 (manuscript with track changes):
*The observed data in Fig. 9a is highly variable compared to the simulations, but this can be related to the more complex topography in reality compared to the smooth model topography. Furthermore, events such as avalanches are not represented in the model.*

> R2-23: L381 The link the authors attribute between the intercept in their linear regression and the compaction is very unclear, please explain.

We have rewritten this paragraph and it now reads:
*The relation suggests that for every 0.01 m of simulated snow distribution 0.011 m of snow redistribution is observed, or in other words, the model underestimates the amount of snow redistribution by only 9.1%. We assume that the compaction rate over the snow pack in the period of 15 hours over the study area is constant and thus the 0.064 m*

*in Equation 11 is related to the compaction of the snow pack. This amount of compaction is in the range of the compaction that we found for a different winter season (between 0.018 m and 0.071 m of the total snowpack decrease of 0.079 m). Therefore, we assume that the average compaction rate of 0.064 m over 15 hours during this study period is realistic. The observed data in Fig. 9a is highly variable compared to the simulations, but this can be related to the more complex topography in reality compared to the smooth model topography. Furthermore, events such as avalanches are not represented in the model. Likewise, the amount of compaction is not absolutely constant over the study area, as this also depends on the snow depth and the weight of overburden layers, and to a minor extent to the wind speeds. However, we assume that variability in compaction is low relative to the effects of snow drift and therefore assume it to be constant.*

R2-24: L389 This results does not only inform on biases but also in on the realism of the spatial pattern of snow redistribution.

Changed.

R2-25: L389 «we have to keep in mind that the TLS data still includes the snowpack compaction». Both observations and simulations include compaction. The compaction of new snow highly prevails after a snowfall event compared to compaction of old snow. Therefore, in this situation, the argument of the authors in their response that initial snow was initialized to a fixed density is not sufficient to consider that simulations are not able to reproduce snow depth change due to compaction. This also has to be considered in the discussion (L434-436), compaction of new snow should prevail in this case.

We agree that the model is able to reproduce snow depth change due to compaction, but this only works if the model is initialized with a realistic snow pack, i.e. fresh layers of snow on layers that are already compacted. This was not possible in our case study due to the high computational costs. Also, we can derive the amount of snow at the glacier with DEM differencing of TLS scans between October 2020 and February 2021, but we do not know any of the physical properties of the snow pack, such as surface temperature or density, which makes a realistic initialization also not viable. We added to L436:
*Also, the amount of snow at the glacier can be derived with DEM differencing of TLS scans between October 2020 and February 2021, but any of the physical properties of the snow pack, such as surface temperature or density remain illusive, which makes a realistic initialization also not viable.*

R2-26: L390-391 «Adding the domain-wide average of the snow compaction rate we found in Fig. 9a leads to inconsistencies in the observational data set; therefore, we omit this step.» As compaction is already simulated by the model, why would the authors want to add again an extra-compaction. Please clarify this unclear statement.

We meant that an average of the obtained compaction of Fig. 9a cannot be subtracted from the observational data set, therefore we omit this step. We do not talk about the model results here. We re-wrote the sentence:
*Adding the spatial average of the snow compaction rate from Fig. 9a to the observational data set leads to inconsistencies; therefore, we omit this step.*

R2-27: L411-412 «The simulated snow redistribution is realistic in terms of spatial structure and magnitude». Is the spatial structure of snow redistribution really realistic? This is far from obvious from Fig 9a. Can you be more accurate and/or mention from which Figure this conclusion is obtained?

We re-wrote the paragraph:
*However, the processes at smaller scales are smoothed out, which is due to the horizontal resolution of 48 m and the smoothed model topography restricted by numerical stability. The model topography limits the slope angles to a maximum of 35°, and thus the model topography clearly deviates from real topography. In agreement with the TLS acquisitions, the simulations show that snow is eroded mostly at the ridges and that the snowpack at the glacier is sheltered and less affected by snow erosion.*

R2-28: L422-423 «Its simplicity compared to fully coupled atmospheric and snow models» Beyond the difference in the snow scheme complexity, could you discuss more accurately which processes are not coupled in this system while they are in CryoWRF and Meso-NH and the possible implication of this uncoupling? I guess there is no interaction between the snow transport module and the water content of the lowest levels of

> the atmospheric models? Also, what is the advantage of using this system compared to completely uncoupled
> system as can be found in hydrological systems (e.g., Marsh et al., 2020; Quéno et al., 2023; Baron et al., 2023)
> It is important to explain that as the disadvantage is rather clear (L429 : «The computationally expensive
> LES cannot be run with a long spin-up time to initialise the snowpack correctly.»).

Our snow drift module within the WRF model is able to provide feedback to the atmosphere: After the calculation of the snow sublimation due to snowdrift, the energy and mass fluxes due to sublimation then modify the specific humidity and air temperature of the atmosphere aloft. This feedback can be switched on or off in the model namelist; in our case, the simulation was ran with full sublimation feedback. Test runs without sublimation suggested that the impact of sublimation on the atmosphere structure aloft was, however, negligibly small. We added a paragraph on the advantages and disadvantages of our approach compared to other models in the manuscript:

*One of the advantages of the presented snow drift module in WRF is its simplicity compared to fully coupled atmospheric and snow models (Vionnet et al., 2013; Sharma et al., 2023), because our snow drift scheme are embedded within the established modules of the WRF modelling system. However, coupling to grain-scale snow models(Vionnet et al., 2013; Sharma et al., 2023) can, of course, provide more detailed information on snowpack evolution and full feedback (fluxes, temperature, humidity) between the atmosphere and the snowpack is possible. In our setup, the feedback of the atmosphere by the snow drift module consists of the impact of snow sublimation on the temperature and special humidity of the atmosphere aloft (Saigger et al., 2023). Furthermore, employing a full physics-based atmospheric model at high resolution provides high-resolution input data for the land surface model. This poses an advantage compared to completely uncoupled hydrological systems (e.g., Marsh et al., 2020; Quéno et al., 2023; Baron et al., 2023), which rely on input from downscaled data, which can be also challenging over complex topography.*

> R2-29: Finally, the discussion is clearly missing an analysis of the reasons for discrepancies between simulated
> and observed spatial patterns of snow depth changes, and perspectives to go beyond pixel-to-pixel evaluations
> in the evaluation of snow transport models.

Thank you for this remark. We added a new paragraph to the discussion:

[revised manuscript text omitted]

---

## Author Response (AR4)

**Response to Editor**

Investigating wind-driven Snow Redistribution Processes over an Alpine Glacier with
high-resolution Terrestrial Laser Scans and Large-eddy Simulations

Annelies Voordendag, Brigitta Goger, Rainer Prinz,
Tobias Sauter, Thomas Mölg, Manuel Saigger and Georg Kaser

January 4, 2024

Dear Dr. Helbig,

Thank you for the two comments on our manuscript. We added our topography smoothing strategy in the model description section (lines 152-158). Furthermore, we added the density distributions to the scatter plot of Figure 9 and added an additional explanation in lines 393-397.

Best regards,
Annelies Voordendag, Brigitta Goger, and co-authors